# Enhancing GUI Agent with Uncertainty-Aware Self-Trained Evaluator

**Gongwei Chen, Lirong Jie, Lexiao Zou, Weili Guan,**∗ **Miao Zhang,**∗ **Liqiang Nie**
Harbin Institute of Technology, Shenzhen
{chengongwei,zhangmiao,guanweili,nieliqiang}@hit.edu.cn
https://github.com/JL181818/URST

## Abstract

Benefiting from the availability of extensive navigation trajectories, both manually and automatically annotated, current graphical user interface (GUI) agents have achieved remarkable advancements in performance. However, these annotated datasets often contain substantial noise, which impedes effective agent training and underscores the necessity for rigorous trajectory quality assessment. In contrast to existing prompting-based evaluators that rely on proprietary multimodal large language models (MLLMs), we propose an Uncertainty-aware Reinforced Self-Training (URST) framework to train lightweight MLLMs for efficient and reliable trajectory evaluation. URST iteratively fine-tunes MLLMs using their own generated thoughts and judgments to enable self-improvement, while its uncertainty-aware sampling strategy ensures the selection of the most informative training examples. To further enhance reasoning and judgment capabilities, we propose a simplified group policy optimization approach that effectively leverages diverse positive and negative samples for evaluator learning. Our evaluator demonstrates superior judgment performance across both in-domain and out-of-domain datasets. When used to filter navigation datasets, it consistently leads to performance improvements in training GUI agents.

## 1 Introduction

With the powerful language understanding and generation abilities, (multimodal) large language model-based agents [53, 62, 7, 28, 38, 45] can perform more effectively across diverse and complex tasks, demonstrating greater generality and adaptability. In a graphical user interface (GUI) scenario, the paradigm for building agents has shifted from framework-based [42, 57, 1] to native agents [6, 14, 20, 31]. In the early stage, many works [42, 57, 1, 61, 16] leverage the advanced foundation models by designing task-specific workflows and optimizing prompts to construct framework-based agents.

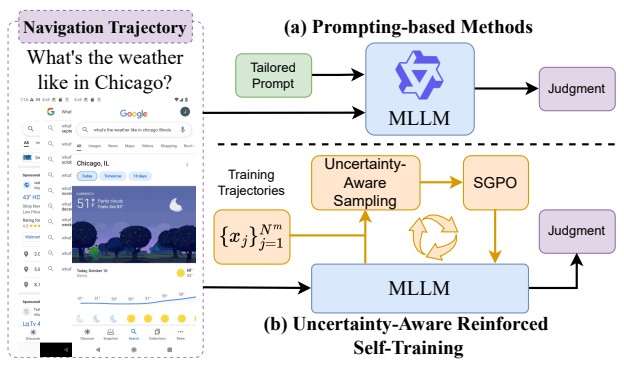

Figure 1: Comparison of uncertainty-aware reinforced self-training (URST) method with prompting-based methods. Our URST iteratively fine-tunes lightweight MLLMs with self-generated thoughts via reinforcement learning.

---

∗Corresponding authors

39th Conference on Neural Information Processing Systems (NeurIPS 2025).

The human-defined workflows make the agents hard to adapt to continuously evolving environments [31]. A promising paradigm is to build native agents by learning GUI navigation tasks in an end-to-end manner, ensuring the scalability and adaptability of agents.

The main challenge of building native agents is the requirement for large-scale GUI navigation data. Recently, a growing number of navigation trajectory datasets [34, 59, 47, 2, 5, 24] have emerged, collected through various schemes ranging from manual to semi-automated and fully automated approaches. Although these datasets have driven significant progress in native agent development, their inherent noise and quality issues continue to hinder further improvement [59, 27, 41]. These data deficiencies demand automated evaluation methods that can act as a quality filter. Current autonomous trajectory evaluation approaches [27, 50, 51, 13] mainly rely on advanced proprietary multimodal large language models (such as GPT-4V, QwenVL-Max, Gemini, etc.) with tailored prompting and continuous API calls. These prompting-based evaluators suffer from two main limitations: (1) the significant domain gap between general and GUI scenarios; (2) the substantial computational cost and delay associated with proprietary MLLM API calling.

To address these challenges, we devise an uncertainty-aware reinforced self-training (URST) method to train an open-sourced lightweight MLLM as an efficient trajectory evaluator. In contrast to supervised fine-tuning on a well-constructed dataset, we explore a self-training strategy that eliminates the need for thought annotations while still enhancing the model's reasoning and judgment capabilities. Our URST method iteratively fine-tunes the model on self-generated data, achieving model self-improvement. In each iteration, our method first performs sampling-based data collection, then fine-tunes the model on newly generated data. In contrast to existing self-training methods [56, 12, 39], we propose a novel uncertainty-aware sampling method to prevent the inclusion of easy samples in the generated dataset. To facilitate efficient learning with diverse positive and negative samples, we devise a simplified group policy optimization method to replace the rejection sampling fine-tuning. To mitigate the difficulty bias, our SGPO eliminates the standard deviation normalization and equally treats each training input, enabling an accurate measurement of sample advantage.

The experiments reveal superior performances of URST in both in-domain (an average gain of $2.45\%$) and out-of-domain (an average gain of $2.89\%$) datasets, compared to the state-of-the-art prompting-based methods. This comparison suggests that domain-specific adaptation through self-training can outweigh the general reasoning power of large foundation models. Notably, evaluators ($84.13\%$) trained with self-generated synthetic data exhibit significantly larger performance than those ($79.01\%$) trained on data from proprietary MLLMs, highlighting the potential of self-generated data. We also implement some self-training works and compare them with URST. The results show that URST outperforms these methods by a margin of $6\%$, thanks to advanced sampling and policy optimization techniques. We also perform GUI agent training with the self-trained evaluator and obtain consistent performance improvements on two navigation datasets. Overall, our findings suggest reinforced self-training as a promising approach to training powerful GUI evaluators.

The key contributions of this work are:

- We introduce URST that enables learning from self-generated data for training efficient MLLM-based evaluators, which is the first fine-tuning work for GUI trajectory evaluation.

- We devise a novel uncertainty-aware sampling method for generating more valuable samples for effective fine-tuning. Meanwhile, simplified group policy optimization is proposed to leverage positive and negative samples for reasoning and judgment enhancement.

- Extensive experiments demonstrate the effectiveness of our URST on two automatically or manually collected datasets. The evaluator trained with URST achieves superior performances in judgment accuracy and shows a positive effect on navigation performance.

## 2   Related Works

**GUI Agents.**   Recently, multimodal large language models (MLLMs) [43, 21, 3, 37, 9, 17] have significantly advanced agent research by enabling richer perception and reasoning across various domains such as gaming environments [45, 18, 19], and GUI interaction [31, 48]. GUI Agents can be categorized into Agent Frameworks and Native Agent Models based on whether they fine-tune the base model on the GUI domain [31]. Agent Framework systems [60, 46, 44, 1] leverage the general understanding and reasoning capabilities of advanced MLLMs. These works enhance the flexibility

of task execution by designing task-specific modules such as planning, execution, memory, reflection, etc., and optimizing prompts for each component. However, agent frameworks depend on manually encoding GUI domain knowledge through custom prompts, external scripts, or tool-usage heuristics, which inherently limit their adaptability to evolving GUI scenarios without human expert involvement. Recent advancements [47, 4, 31, 11, 20] have shifted towards constructing large-scale GUI operating trajectories for fine-tuning MLLMs, thereby enabling end-to-end perception, planning, and execution natively within the GUI domain. These native agent models reduce the need for human-engineered workflows by utilizing either manually annotated or automatically collected trajectories, providing the potential for self-evolution as environments change. In contrast, our work tries to build an evaluator to measure trajectory quality, remove ineffective samples automatically, and ultimately enhance the performance of GUI agents.

**Autonomous Evaluation by LLMs.** Recently, LLM-as-a-judge approaches have been proposed to evaluate the correctness of the trajectory. AutoEval [27] prompts a general advanced MLLM to evaluate task completion with the last screenshot. WebVoyager [13] further inputs all screenshots for judgment. Some works [50, 41, 40] also incorporate a general LLM to evaluate sample quality during dataset construction. Webjudge [51] proposes an evaluation pipeline with key screenshots identification to reduce token consumption. These approaches rely on general proprietary LLMs, which lack domain-specific knowledge tailored to GUI environments. In contrast, we advocate transitioning from prompting-based to fine-tuned evaluators, facilitating domain-specific evaluation via model self-improvement.

**Reinforcement Learning with LLMs.** Reinforcement Learning (RL) has been demonstrated to significantly enhance model performance during the post-training phase of LLMs Early works utilized RL to align LLM outputs with human preferences [55] through algorithms such as Proximal Policy Optimization [35] and Direct Preference Optimization [32]. More recently, Deepseek [36, 10] proposes group relative policy optimization (GRPO) by introducing group-level advantage estimation, and substantially augments the reasoning capabilities of LLMs. Following this pioneering work, a growing body of studies [26, 58, 22, 29] have applied GRPO across various domains, consistently reporting notable performance gains. Our work leverages group-level policy optimization for enhancing the trajectory evaluator's reasoning ability in self-evolved training. To the best of our knowledge, this is the first study to employ reinforcement learning for evaluating GUI trajectories.

## 3 Method

### 3.1 Problem Definition

GUI navigation task requires an agent to predict a sequence of actions and interact with an environment based on task instructions. Here, we focus on vision-based GUI navigation offline datasets, where each sample is composed of a task instruction $I$, a sequence of state-action pairs $S = \{s_0, a_0, \ldots, s_T, a_T\}$. In this work, $s_t$ represents a screenshot observation instead of a structural text description (like A11y tree). Given a navigation sample, the trajectory evaluator $\mathcal{E}$ produces a chain-of-thought (CoT) $\mathbf{c}$, and a judgment $y$, which can be formulated as follows:

$$(\mathbf{c}, y) = \mathcal{E}(x), \ x = \{I, S\}. \tag{1}$$

Due to the introduction of CoT reasoning, we adopt a language modeling objective to train the evaluator instead of a binary classification objective. The CoT reasoning provides abundant information about task intention, state content, and task progress to make a solid judgment.

The dominant paradigm of constructing automatic evaluators for GUI Agent is adapting a proprietary MLLM with tailored prompts, termed prompting-based evaluator. This paradigm is quite expensive and relies on API calls to proprietary models. Another direction is to fine-tune an open-sourced MLLM with the training samples annotated by advanced proprietary MLLMs for low-cost local deployment. However, this supervised fine-tuning method may be constrained by the limited capability of proprietary MLLMs in domain-specific tasks. In this work, we devise a novel uncertainty-aware reinforced self-training method to fine-tune open-sourced MLLMs as trajectory evaluators, enabling model self-improvement through their own outputs. The whole procedure is shown in Algorithm 1.

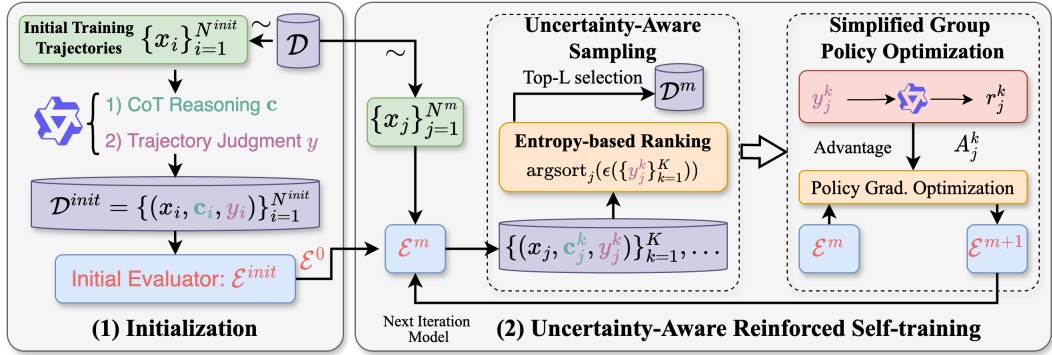

Figure 2: Overview of our uncertainty-aware reinforced self-training (URST) framework. We begin by using an initial training set to adapt a general MLLM to the trajectory evaluation domain. Subsequently, URST iteratively fine-tune the MLLM with its own thoughts and judgments, guided by uncertainty-aware sampling and simplified group policy optimization.

## 3.2 Uncertainty-aware Reinforced Self-Training

Our approach first assumes access to an initial MLLM-based evaluator $\mathcal{E}^{init}$, a set of training inputs $\mathcal{D}$, and a reward model $\mathcal{R}$ to judge the correctness of the outputs. Given a training input $x = \{I, S\}$, the MLLM-based evaluator is expected to generate (1) a CoT reasoning $\mathbf{c}$, followed by (2) a final judgment $y$ to the given input. We require a reward model to verify the correctness of the final judgments, rather than assessing the internal CoT reasoning process. Specifically, a proprietary MLLM is adapted to verify the judgment as the reward model. Our method iteratively samples multiple training inputs from $\mathcal{D}$ and outputs from the current evaluator $\mathcal{E}$, then constructs training datasets to fine-tune the evaluator via simplified group policy optimization.

### 3.2.1 Initialization

As the trajectory evaluation is not a common task in MLLM pre-training or instruction-tuning, current open-sourced MLLMs, especially small ones, struggle with understanding high-level navigation instructions and complex state-action pairs. We construct and collect a small amount of trajectory evaluation data $\mathcal{D}^{init} = \{(x_i, \mathbf{c}_i, y_i)\}_{i=1}^{N^{init}}$ to fine-tune an open-sourced MLLM as the initial evaluator. To collect such data, we directly prompt an advanced proprietary MLLM to generate trajectory judgments with detailed CoT reasoning.

Given the initial trajectory evaluation dataset $\mathcal{D}^{init}$, we fine-tune an instruction-tuned MLLM $\pi_B$ as the initial evaluator by minimizing the negative log likelihood loss:

$$\mathcal{L}_{SFT} = -\mathbb{E}_{(x_i, \mathbf{c}_i, y_i) \sim \mathcal{D}^{init}} \left[ \log \mathcal{E}^{init}(\mathbf{c}_i, y_i | x_i) \right]. \tag{2}$$

After the initialization, our MLLM-based evaluator acquires a basic ability to assess unseen trajectory data. We start from this initial evaluator and devise an uncertainty-aware sampling method to expand the training data and use iterative reinforced self-training to develop a more powerful evaluator.

### 3.2.2 Uncertainty-Aware Sampling

Given a set of training inputs $\mathcal{D}$, a MLLM-based evaluator $\mathcal{E}^m$, we sample a subset of training inputs $\{x_j\}_{j=1}^{N^m}$ in the $m$-th iteration, and generate K output sequences $\{(\mathbf{c}_j^k, y_j^k)\}_{k=1}^K$ for each training input $x_j$ based on the current evaluator. Specifically, by setting the temperature to a value greater than 0, the MLLM-based evaluator can produce different outputs for the same input through $K$ repeated inference. Based on the generated dataset $\mathcal{D}^m = \{(x_j, \mathbf{c}_j^k, y_j^k) \mid j \in [N^m], k \in [K]\}$, a reward model will score these samples by verifying the correctness of their judgments $y$. In our experiments, the reward score simply corresponds to $\tilde{r}_j^k = 1$ if $y_j^k = y_g$, and 0 otherwise, $y_g$ is the ground truth annotated by the reward model.

The self-generated dataset $\mathcal{D}^m$ with the rewards can be directly used to fine-tune the current evaluator to yield a new evaluator for the next iteration. However, we find the random sampling of inputs is

---

**Algorithm 1** Uncertainty-aware Reinforced Self-Training

---

**Input:** an instruction-tuned MLLM $\pi_B$, a set of training inputs $\mathcal{D}$, a reward model.
// Initialization
Generate the initial dataset $\mathcal{D}^{init}$ by using a proprietary MLLM: $\mathcal{D}^{init} = \{(x_i, \mathbf{c}_i, y_i)\}_{i=1}^{N^{init}}$
Fine-tune $\pi_B$ to obtain the initial evaluator $\mathcal{E}^{init}$     $\triangleright$ Cold start by supervised fine-tuning
$\mathcal{E}^0 = \mathcal{E}^{init}$
**for** $m = 0$ **to** $M - 1$ **do**
    // Uncertainty-aware Sampling
    Generate dataset $\mathcal{D}^m$ by uncertainty-aware sampling: $\mathcal{D}^m = \{(x_j, \mathbf{c}_j^k, y_j^k) \mid j \in [L], k \in [K]\}$ s.t. $x_j \sim \mathcal{D}, (\mathbf{c}_j^k, y_j^k) \sim \mathcal{E}^m$.
    Annotate $\mathcal{D}^m$ with the reward $r_j^k$.
    // Simplified Group Policy Optimization
    Obtain $\mathcal{E}^{m+1}$ by minimizing the SGPO objective in equation 5.
**end for**
**Output:** The final evaluator $\mathcal{E}^M$

---

inefficient due to the inclusion of less informative samples. To address this issue, we propose an uncertainty-aware sampling method to choose the hard samples in each iteration. Our method ranks the training input based on the prediction uncertainty of the evaluator. We use the entropy of multiple judgments over the same input to measure the prediction uncertainty. Given the generated outputs $\{y_j^k|_{j=1}^K\}$, the entropy $\epsilon_j$ of the training input $j$ is computed as follows:

$$
\begin{aligned}
\epsilon_j &= -p_{yes} \log p_{yes} - (1 - p_{yes}) \log(1 - p_{yes}), \\
p_{yes} &= \frac{1}{K} \sum_{k=1}^{K} \mathbb{I}(y_j^k = \text{Yes}).
\end{aligned}
\tag{3}
$$

The value range of judgment $y$ is $\{\text{Yes}, \text{No}\}$. $\mathbb{I}(\cdot)$ is an indicator function. We select Top-$L$ samples with the highest entropy values to construct the generated dataset. Higher entropy means that the evaluator can't make a solid decision on the training input, suggesting its value to learn. Compared to random sampling in existing self-training methods, our uncertainty-aware sampling focuses on the most valuable samples in each iteration, and efficiently expands the training dataset to ensure the diversity and richness of the final dataset.

### 3.2.3 Iterative Reinforced Self-Training

In the $m$-th iteration, we use the newly generated dataset $\mathcal{D}^m$ from the uncertainty-aware sampling phase to fine-tune the current evaluator, yielding a new evaluator $\mathcal{E}^{m+1}$. Once the evaluator is improved, a new dataset of high-quality samples can be generated once again. The data generation and training steps can be iteratively performed M times, resulting in a final evaluator $\mathcal{E}^M$.

Existing reinforced self-training methods [56, 15, 12, 39] use rejection sampling fine-tuning to iteratively train the model. This fine-tuning scheme uses reward values to filter the generated samples and keep positive outputs for training. It has one main limitation: the exclusion of negative outputs. Incorrect outputs can also contain valuable information that helps the model identify error patterns in generations. To address this issue, we introduce Simplified Group Policy Optimization (SGPO) in our iterative reinforced self-training framework.

### 3.2.4 Simplified Group Policy Optimization

The recently proposed group relative policy optimization (GRPO) [36] compares groups of candidate responses directly, mining discriminative positive and negative outputs within the same input. The group-level optimization introduces comprehensive contrastive signals for efficient learning. Inspired by GRPO and some recent variants [8, 23, 54], we design simplified group policy optimization (SGPO) for better group-level contrastive learning. We first design two types of rewards, format reward and judgment reward.

- **Format rewards** $\hat{r}$**:** We employ a format reward model that enforces the model to put its CoT reasoning process between "`<think>`" and "`</think>`" tags.
- **Judgment rewards** $\tilde{r}$**:** The judgment reward model will score the samples by verifying the correctness of their judgments $y$. We use the rewards in the uncertainty-aware sampling phase as judgment rewards.

SGPO computes the relative quality of $K$ different outputs $\{(\mathbf{c}_j^k, y_j^k)\}_{k=1}^K$ based on the summation of two rewards $r = \hat{r} + \tilde{r}$. We can directly measure the relative quality $A_j^k$ as follows:

$$A_j^k = r_j^k - \text{mean}(\{r_j^1, \ldots, r_j^K\}). \tag{4}$$

Note that our SGPO eliminates the standard variance normalization in the original GRPO. This normalization can help to mitigate the influence of extreme reward values [8], but may introduce input-level difficulty bias [23]. After our uncertainty-aware sampling, the standard deviations of all output groups fall within a relatively small range ($[0.23, 0.47]$ in the first iteration), indicating the absence of extreme reward values. In this case, the normalization term assigns greater weight to samples with smaller standard deviations, thereby overemphasizing these less informative samples (low standard deviation means low entropy). Thus, removing normalization can yield positive gains by treating each training input equally.

Then SGPO optimizes the evaluator $\mathcal{E}$ by minimizing the following objective:

$$\mathcal{L}_{SGPO} = -\mathbb{E}\left[\frac{1}{K}\sum_{k=1}^K \left(\min\left(\mathbf{h}_j^k A_j^k, \text{clip}(\mathbf{h}_j^k, 1-\tau, 1+\tau)A_j^k\right) - \beta\mathbb{D}_{KL}\right)\right], \tag{5}$$

where $\tau$ and $\beta$ are hyper-parameters, and $\mathbf{h}_j^k = \frac{\mathcal{E}^m(\mathbf{c}_j^k, y_j^k|x_j)}{\mathcal{E}_{old}^m(\mathbf{c}_j^k, y_j^k|x_j)}$. $\mathbb{D}_{KL}$ is adopted to regularize the model, preventing excessive deviation from the initial evaluator, which is estimated as in [10].

## 4 Experiments

### 4.1 Datasets

Table 1: Comparison of our URST with prompting-based and fine-tuned evaluators on three test sets, AITW-ID-traj, AITW-OOD-traj, and AW-OOD-traj. "Iter." denotes iterative training process. We adopt QwenVL-Max and Qwen2.5VL-3B as the base MLLM for prompting-based and fine-tuning-based methods, respectively. "URST*" use Qwen2VL-2B as the base MLLM. "AutoEval*" uses the last two screenshots as input, which is in line with the fine-tuning methods.

| Method | Iter. | AITW-ID-traj Acc. | AITW-ID-traj F1 | AITW-OOD-traj Acc. | AITW-OOD-traj F1 | AW-OOD-traj Acc. | AW-OOD-traj F1 | Overall Acc. | Overall F1 |
|---|---|---|---|---|---|---|---|---|---|
| \multicolumn{10}{c}{Prompting-based Methods} |
| AgentTrek [50] | ✗ | 84.17 | 87.58 | 90.00 | 70.00 | 73.54 | 69.74 | 80.56 | 76.80 |
| WebVoyager [13] | ✗ | 85.83 | 88.74 | 88.33 | 65.00 | 81.61 | 77.10 | 84.45 | 80.54 |
| WebJudge [52] | ✗ | 82.50 | 85.91 | 85.71 | 66.67 | 77.58 | 70.59 | 79.43 | 77.64 |
| AutoEval [27] | ✗ | 83.33 | 86.30 | 94.16 | 80.00 | 80.72 | 73.94 | 84.88 | 79.77 |
| AutoEval* | ✗ | 85.00 | 88.00 | 94.16 | 80.00 | **82.06** | 75.90 | 85.95 | 81.48 |
| \multicolumn{10}{c}{Fine-tuning-based Methods} |
| SFT | ✗ | 84.17 | 87.42 | 94.17 | 78.79 | 77.58 | 71.91 | 83.59 | 79.01 |
| STaR [56] | ✔ | 81.67 | 84.51 | **95.00** | 81.25 | 78.03 | 68.39 | 83.37 | 76.60 |
| ReST$^{EM}$ [39] | ✔ | 84.17 | 87.25 | 91.67 | 72.22 | 79.82 | 70.59 | 84.02 | 78.11 |
| URST* | ✔ | 82.50 | 86.79 | 89.17 | 71.11 | 81.61 | 78.53 | 83.80 | 81.01 |
| URST | ✔ | **87.50** | **90.45** | 94.16 | **82.05** | 81.61 | **79.60** | **86.39** | **84.13** |

We collect and construct a training set and three test sets for GUI trajectory evaluation. The training set is built on a subset of Android-in-the-Wild (AITW) datasets. As analyzed in [27], about 36% of the human demonstrations in this dataset are actually incorrect. We randomly sample 1500

trajectories from AITW training set, and use Qwen-VL-Max to generate the thoughts and judgments for supervised fine-tuning. In the self-training setting, we only sample 300 trajectories with the thoughts and judgments generated from Qwen-VL-Max as the initial training set.

**AITW-ID-traj** and **AITW-OOD-traj** are in-domain and out-of-domain test sets built on AITW dataset. Each of these two test sets contains 120 tasks and was manually annotated in [27]. AITW-ID-traj and AITW-OOD-traj share the task goals, but have different trajectory distributions. Following OS-genesis [41], we also collect some agent-executed trajectories from an online environment, AndroidWorld. After manually annotation and filtering, we keep 223 trajectories and obtain a new out-of-domain test set, **AW-OOD-traj**. The implementation details can be found in Appendix.

## 4.2 Main Results

To comprehensively assess the advantages of our method, we compare it with prompting-based works, AgentTrek [50], WebYoyager [13] , Webjudge [52], and AutoEval [27], following the settings in the original papers. Meanwhile, we also implement some representative self-training methods as fine-tuning-based baselines (i.e., STaR [56], ReST$^{EM}$ [39]), which are originally designed for LLM post-training instead of trajectory evaluator training. The implementation details of these self-training methods can be found in Appendix.

The results in Table 1 show that our URST consistently achieves the highest F1 score with an average of $84.13\%$. Among the prompting-based methods, WebVoyager and AutoEval demonstrate superior performances across all datasets. WebVoyager uses all screenshots of each trajectory as input, achieving better performance than methods with incomplete screenshots, but it suffers from significant token overload. Following the setting of fine-tuning methods, we augment AutoEval with the last two screenshots, resulting in an advanced AutoEval* with the best performance among the prompting-based works. Despite the success of these methods, URST still outperforms them by leveraging a reasoning-enhanced self-training strategy.

To fully showcase its advantages, we compare our URST with supervised fine-tuning (SFT) and various self-training methods. SFT uses the thoughts and judgments generated by an advanced proprietary MLLM (Qwen-VL-Max), which can be treated as Distillation. By inheriting the reasoning patterns of a larger base MLLM, the smaller model Qwen2.5VL-3B demonstrates comparable evaluation capability ($81.48\%$ vs. $79.01\%$). Self-training methods try to use self-generated thoughts and judgments to achieve model self-improvement. We find current self-training approaches, STaR and ReST$^{EM}$ indicate inferior results to SFT with externally generated thoughts, which can be attributed to the less valuable training examples and limited reasoning enhancement ability of rejection fine-tuning. In contrast, URST achieves the highest results across all metrics, highlighting the value of uncertainty-aware sampling and group-level policy optimization.

## 4.3 Ablation Studies

We conduct the ablation studies for our URST with Qwen2.5VL-3B to examine the effectiveness of uncertainty-aware sampling and group relative policy optimization. As shown in Table 2, eliminating uncertainty-aware sampling and SGPO decreases the overall performance by $6.00\%$ and $9.49\%$, demonstrating the significant value of these two key components. Without uncertainty-aware sampling, our method cannot identify the most valuable training data, and is prone to overfitting due to excessive fine-tuning ($-3.73\%$). SGPO can mitigate this overfitting problem and maintain its generalization on OOD evaluation datasets ($0.0\%$ on AITW-OOD-traj and $+1.46\%$ on AW-OOD-traj) compared to SFT. On the other hand, without SGPO, uncertainty-aware sampling can maintain the out-of-domain performance (from $78.95\%$ to $78.79\%$ on AITW-OOD-traj) under the same environment, but impair the out-of-domain capability (from $71.91\%$ to $63.22\%$ on AW-OOD-traj) across different environments, compared to SFT. Incorporating advantage normalization consistently decreases the performance, underscoring the importance of accurately measuring sample advantages.

**Impact of multiple iterations.** As shown in Figure 3, our results show that multiple iterations can consistently enhance the performance of the evaluator, especially on F1 scores. The performance gains across three iterations are $0.65\%/1.55\%$, $1.50\%/3.42\%$, $0.44\%/1.28\%$ in terms of accuracy and F1 score, respectively. Iteration 1 leads to the biggest performance increase, while performing

Table 2: Ablation studies for different components of the proposed URST. We report the F1 score.

| Method | AITW-ID-traj | AITW-OOD-traj | AW-OOD-traj | Overall |
|---|---|---|---|---|
| **URST** | 90.45 | 82.05 | 79.60 | 84.13 |
| w/o uncertainty-aware sampling | 83.69 | 78.79 | 73.37 | 78.13 |
| w/o SGPO | 85.33 | 78.95 | 63.22 | 74.64 |
| w/o normlization removing | 89.61 | 78.95 | 77.95 | 82.69 |
| SFT | 87.42 | 78.79 | 71.91 | 79.01 |

more iterations slightly reduces the gain effect due to the relatively low quality of the newly sampled training data, as shown in Figure 4.

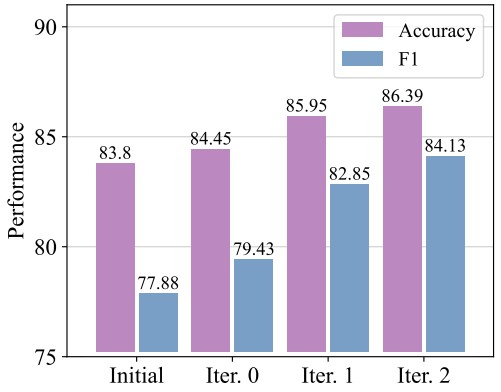

Figure 3: Analysis of the iteration number $M$. Our URST consistently boosts the performance of the evaluator.

Figure 4: The change of entropy distributions over various iterations. We present entropy distributions of different sampling methods, random and uncertainty-aware sampling.

**Impact of the number $K$ of generated outputs per trajectory.** Table 3 reveals how training with different numbers of $K$ affects evaluator performance. Generally, increasing the sampling number $K$ consistently enhances the evaluator's performance across in-domain and out-of-domain datasets. When increasing $K$ from 4 to 8, F1 score of AITW-OOD-traj has the largest performance gain of 12.5%, indicating the significant OOD generalization effect from comprehensive group-level contrastive learning. However, further enlarging $K$ to 16 led to a performance decrease. We speculate that a large sampling number forces the model to generate overly homogeneous responses, leading to overfitting. Overall, $K = 8$ shows the best F1 scores across all trajectory evaluation datasets.

Table 3: Analysis of the sampling number $K$. We perform the comparison experiments under the setting of one iteration.

| Sampling Number $K$ | AITW-ID-traj | AITW-OOD-traj | AW-OOD-traj | Overall |
|---|---|---|---|---|
| 4 | 84.51 | 75.00 | 72.19 | 77.55 |
| 8 | 86.11 | 87.50 | 72.41 | 79.43 |
| 16 | 84.72 | 87.50 | 71.76 | 78.61 |

**Impact of the entropy.** Our uncertainty-aware sampling uses sampling-based entropy to measure the uncertainty of the evaluator, which is inspired by self-consistency methods in LLM [63]. Besides the proposed metric, we also devise an alternative metric, prediction-based entropy, which uses greedy decoding to generate a single judgment of each training input, and computes the probability entropy of the judgment prediction. In the Table 4, we compare these two metrics with our URST. Sampling-based entropy consistently outperforms prediction-based entropy across all test sets. Our sampling-based entropy uses temperature sampling and entropy of multiple outputs to measure the uncertainty of the evaluator, which could explore various reasoning paths for answer generation and provide a reliable assessment of the model's confidence. However, prediction-based entropy uses greedy decoding to generate a single output for entropy computation, and is prone to suffer from the over-confidence issue of LLM [49], thus making less reliable measurement.

Table 4: The comparison of entropy variants in our uncertainty-aware sampling.

| URST | AITW-ID-traj | AITW-OOD-traj | AW-OOD-traj | Overall |
|------|--------------|---------------|-------------|---------|
| w/ prediction-based entropy | 89.47 | 73.68 | 75.68 | 81.07 |
| w/ sampling-based entropy | 90.45 | 82.05 | 79.60 | 84.13 |

## 4.4 The enhancement of GUI Agent

By using an evaluator to filter the incorrect trajectories, we can improve the quality and reliability of the dataset. We choose two GUI navigation datasets to conduct filtered Behavior Cloning training with our trained evaluator. One is AITW dataset [34], which is constructed by human experts. Another is automatically collected OS-Genesis dataset [41]. We first filter the original navigation dataset $\mathcal{D}^o$ by discarding the trajectories identified as failures. This filtering process will produce a new dataset $\mathcal{D}^c$ by using the trained evaluator. The same number of trajectories from the original and filtered datasets is randomly sampled to train GUI Agents.

Table 5: Comparisons of GUI navigation performances with original and filtered datasets. The base MLLM is Qwen2.5VL-3B. We choose AITW and OS-Genesis as the representatives of manually and automatically generated noisy datasets and evaluate them under the settings of standard and high noise. We report the step success rate, average time cost of filtering samples, and cost per 1M tokens.

| Datasets | Time (s) | Cost ($) | AITW | | OS-Genesis | |
|----------|----------|----------|------|------|------------|------|
| | | | Standard | High | Standard | High |
| Original $\mathcal{D}^o$ | 0 | 0 | 63.50 | 20.88 | 46.83 | 46.43 |
| $\mathcal{D}^c$ by AutoEval* | 5.99 | 0.41 | 64.98 | 38.93 | 47.26 | 47.65 |
| $\mathcal{D}^c$ by SFT | 2.17 | 0 | 64.32 | 32.59 | 47.18 | 47.22 |
| $\mathcal{D}^c$ by URST | **2.17** | **0** | **65.02** | **40.41** | **47.66** | **48.26** |

We conducted some comparison experiments by using two settings, Standard-Noise and High-Noise.

- **Standard-Noise**: Discarding all incorrect trajectories, and using all steps in each trajectory for training GUI agents.

- **High-Noise**: Discarding all incorrect trajectories, and using the last 40% of steps in each trajectory for training GUI agents. The underlying rationale is that the majority of incorrect trajectories tend to fail in the last few steps. When tasks are highly challenging or involve unseen environments, the data automatically collected by the agent often contains high levels of noise, meaning that the entire trajectory is wrong from the very beginning.

Based on the Table 5, we can conclude that 1) our URST can consistently outperform other baselines (including AutoEval* and SFT) in filtering noisy datasets. 2) When the noisy level rises, all filters (based on AutoEval*, SFT, and URST) can achieve a greater impact on agent performance, with URST consistently achieving the best result by a significant margin. 3) URST demonstrates a clear overall advantage, delivering optimal outcomes in time cost, expenditure, and performance.

In an online environment, the trajectory evaluator can serve as a reward signal to enhance an existing GUI agent at inference time, using the Reflexion technique [38]. Specifically, a GUI agent first attempts a task, and an external evaluator is used to judge whether its attempt was successful or not. The agent will be prompted to reflect on the failure and retry unsuccessful tasks. We conduct experiments on the most popular online benchmark, AndroidWorld, with M3A agents [33]. The best prompting-based method, AutoEval*, and our URST are used as a reward signal for eliciting reflection. Table 6 shows that URST achieves superior performance, with a 1.8% improvement over AutoEval*, underscoring its clear advantage in enhancing GUI agents.

Table 6: Comparison of online navigation performance of M3A [33] using the Reflexion technique [38] with different evaluators. We report the success rate.

| Methods | AndoridWorld |
|---------|--------------|
| M3A | 39.7 |
| M3A + AutoEval* | 44.8 |
| M3A + URST | 46.6 |

## 4.5 Further Analysis

**The change of entropy distribution over various iterations.** In Figure 4, we illustrate the entropy distributions of different sampling methods across various iterations. The training data sampled by our URST exhibits significantly higher entropy, indicating its high quality. As the number of iterations increases, the entropy of the training examples gradually decreases, due to the improved capability of the evaluator.

**Case Study.** To further illustrate the reasoning capabilities of the proposed method, we present a qualitative case study comparing URST with SFT baseline in trajectory evaluation tasks, as shown in Figure 5. Unlike models that passively imitate annotated thoughts, often leading to shallow reasoning or hallucination issues, our URST maintains high reasoning accuracy by generating its own intermediate thoughts and conducting group-level contrastive training. In out-of-domain scenarios, URST demonstrates greater robustness by accurately leveraging contextual cues and interface semantics.

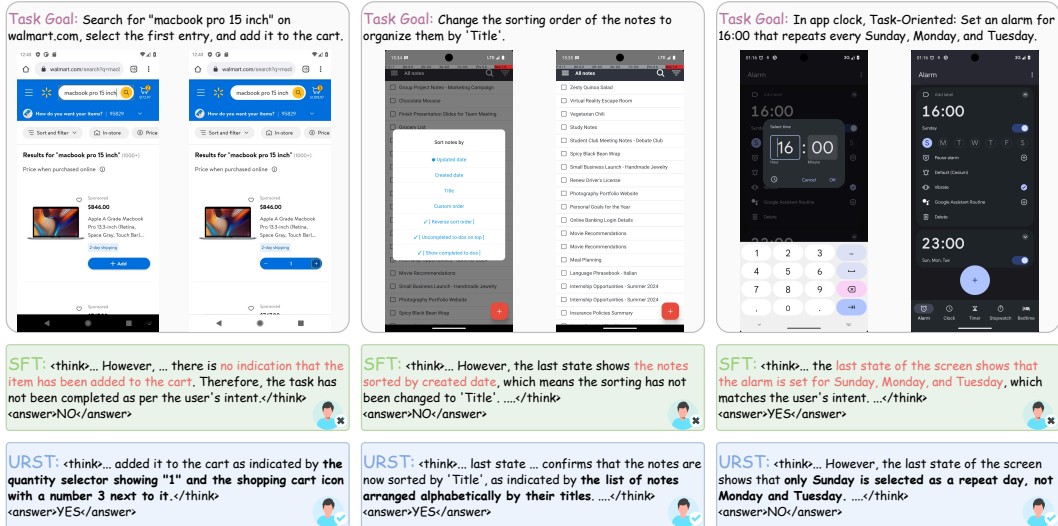

Figure 5: Illustration of some trajectory evaluation tasks with the model-generated reasoning. The left one is an in-domain task, and the others are out-of-domain tasks. URST produces correct judgments through precise reasoning, whereas SFT fails.

## 5 Conclusion

In this work, we propose an uncertainty-aware reinforced self-training (URST) framework to train lightweight MLLMs as efficient GUI trajectory evaluators. URST integrates uncertainty-aware sampling and simplified group policy optimization (SGPO) to effectively identify informative samples and enhance reasoning capabilities. The approach achieves superior performance on both in-domain and out-of-domain evaluation datasets and enhances downstream GUI agent performance by filtering noisy training data. This work offers a scalable and promising approach for building robust GUI trajectory evaluators, paving the way for more adaptive and capable GUI navigation agents.

## Acknowledgements

We thank anonymous reviewers for their constructive feedback. This study is supported by Shenzhen Science and Technology Program (KQTD20240729102207002), National Natural Science Foundation of China (Grant No. U24A20328, 62476071, 62306084, and U23B2051), Shenzhen College Stability Support Plan under Grant GXWD20231128102243003, and Shenzhen Science and Technology Program under Grant ZDSYS20230626091203008 and KJZD20230923115113026, China Postdoctoral Science Foundation (Grant No. 2024M764192).

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

# A  Trajectory Evaluation Datasets

The AITW-ID-traj and AITW-OOD-traj evaluation datasets are directly adopted from the setup introduced in AutoEval [27]. Both datasets are based on the Android-in-the-Wild (AITW) [34] benchmark, which consists of human-executed trajectories collected in Android emulator environments.

The AITW-ID-traj test set includes 120 trajectories randomly sampled from the General, WebShopping, and GoogleApps subsets of the AITW training set (40 episodes from each). These trajectories were originally collected via human demonstrations on an Android emulator, where annotators manually executed task flows. However, the data contains significant noise. Common failure cases include early termination, completing the wrong task, or making mistakes in task parameters. As a result, the correctness of these episodes is not guaranteed. After manual verification, only approximately 64.2% of the selected episodes in AITW-ID-traj were found to be successful.

The AITW-OOD-traj test set shares the same task goals as AITW-ID-traj but differs in the source of trajectories. In this case, the trajectories were generated by CogAgent [14], an autonomous agent interacting with an Android emulator. The emulator was configured using Android Studio's built-in environment to simulate a Pixel 4 device with API level 33. Manual annotation revealed that only 14.2% of the generated trajectories successfully completed the assigned tasks.

Following OS-Genesis [41], we adopt a fully automated data collection process to construct the AW-OOD-traj test set. This set is built using automatically generated task goals and agent-executed trajectories within the AndroidWorld [33] environment. We employ a reverse task synthesis approach to create a diverse range of high-level task goals that align with dynamic environments. The agent from Agent Q [30] is then used to interact with the environment based on these task goals, generating approximately 500 new trajectories. We filter out low-quality trajectories that are too short or based on invalid task goals. After filtering, each remaining trajectory is manually annotated using a major voting scheme, resulting in the final AW-OOD-traj test set containing 223 trajectories. 43.0% of trajectories in this test set are annotated as successful.

Table 7: The statistics of three trajectory evaluation datasets.

| Datasets | #Success | #Total | Success Rate |
|---|---|---|---|
| AITW-ID-traj | 77 | 120 | 64.17% |
| AITW-OOD-traj | 17 | 120 | 14.17% |
| AW-OOD-traj | 96 | 223 | 43.05% |

# B  Experimental Details

## B.1  Implementation Details

All experiments were conducted on 4 NVIDIA A100 40GB GPUs. The key hyperparameters used in our experiments are summarized in Table 8. The model is trained for 2 epochs using 300 samples annotated with Qwen-VL-Max during the initialization SFT stage. For SRPO training, each iteration involves 4 epochs of training on 400 samples sampled via URST. Consequently, after 3 iterations, the total number of training samples amounts to 1500. Both Initialization and subsequent SGPO training are conducted with full-parameter fine-tuning. In each Iteration, the learning rate is warmed up linearly from 0 to 1e-6 across 5 global steps and then reduced to a minimum of 0 using cosine decay. We adopt a $\beta$ value of 0.001, which balances the reward signal and divergence constraint in the policy update. To manage training efficiency and computational cost, the maximum pixel limit for each visual input was set at 802,816. If an input image exceeds this limit, it is cropped and resized while preserving the original aspect ratio. To further enhance memory efficiency and scalability, all experiments are conducted using DeepSpeed's Zero-3 optimization stage, and flash attention is employed to accelerate training. During both the uncertainty-aware sampling and SGPO training, we apply the temperature set to 1.0, top-$k$ sampling with $k = 50$, and nucleus (top-$p$) sampling with $p = 0.9$ to ensure the diversity of the outputs.

An example of the prompt format used during training and inference is illustrated in Figure 9. Specifically, we incorporate the last two states of the trajectory into the input of the MLLM. This

Table 8: Hyperparameter settings used in the experiments.

| Hyperparameter | Value |
|---|---|
| SFT training epoch at initialization stage | 2 |
| SRPO training epoch | 4 |
| sample size per iteration | 400 |
| $\beta$ | 0.001 |
| learning rate | 1e-6 |
| warmup ratio | 0.05 |
| max pixels | 802,816 |
| per device train batch size | 2 |
| DeepSpeed optimization stage | Zero-3 |
| temperature | 1 |
| top-$k$ | 50 |
| top-$p$ | 0.9 |

design enriches the prompt with critical state transition information while avoiding excessive context length and token overhead. Furthermore, the action history is refined to explicitly include both click coordinates and scroll directions, offering a more fine-grained representation of user interactions. These enhancements provide the model with richer transition cues, thereby improving its ability to make informed and accurate predictions.

During the data filtering process for Behavior Cloning training, we leverage our trained evaluator to assess and filter the original navigation dataset $\mathcal{D}^o$, resulting in a cleaned dataset $\mathcal{D}^c$. The retention rate of the trajectories is reported in Table 9. In the subsequent training of the GUI navigation agent, we employ SFT with Qwen2.5-VL-3B for 2 epochs. The prompt templates used during training and inference for the AITW and OS-Genesis datasets are illustrated in Figure 15 and Figure 16.

Table 9: The retention rate of trajectories in two GUI navigation datasets after being filtered by our trained evaluator.

| AITW | | | | OS-Genesis |
|---|---|---|---|---|
| General | WebShop | GoogleApps | Total | |
| 81.25% | 40.20% | 81.85% | 67.77% | 45.14% |

### B.2 The compared methods

To comprehensively assess the advantages of our method, we compare it with prompting-based works, AgentTrek [50], WebVoyager [13] , Webjudge [51], and AutoEval [27], following the settings in the original papers. The prompt templates used for these prompting-based evaluation methods can be found in Figure 10, 11, 12, 13. Following the fine-tuning methods, we also implement an advanced version of AutoEval by providing the last two screenshots, termed AutoEval*. The prompt template used for AutoEval* is illustrated in Figure 14. Meanwhile, we also implement some representative self-training methods as fine-tuning-based baselines, which are originally designed for LLM post-training instead of trajectory evaluator training. Here, we briefly introduce the compared self-training methods.

- **Self-Taught Reasoner (STaR) [56]** employed greedy decoding instead of temperature sampling for data generation, which is restricted to one model-generated response per input during data collection. Here, we adopt temperature sampling to generate responses and choose one of the correct responses as training data.

- **Reinforced Self-Training with Expection-Maximization (ReST$^{EM}$)** [39] is a simplified and advanced version of ReST [12], which decouples data collection and policy optimization in the reinforcement learning pipeline. ReST iteratively conducts self-generated data collection and reward-weighted fine-tuning to achieve self-improvement of the policy. However, it has one main restriction: requiring human-generated outputs, which are unavailable in

our experiments. The main differences of ReST$^{EM}$ from ReST are the exclusion of human-generated outputs and the iteratively fine-tuning on the base model instead of the model from the previous iteration. Here, we set the cut-off threshold for the maximum number of responses per input to 4.

To ensure a fair comparison, we implement all the above methods on a dataset matching the size of the final dataset (1500 training inputs) in our URST method, which includes data from the initial phase and all iterative sampling steps.

## C   More Analyses

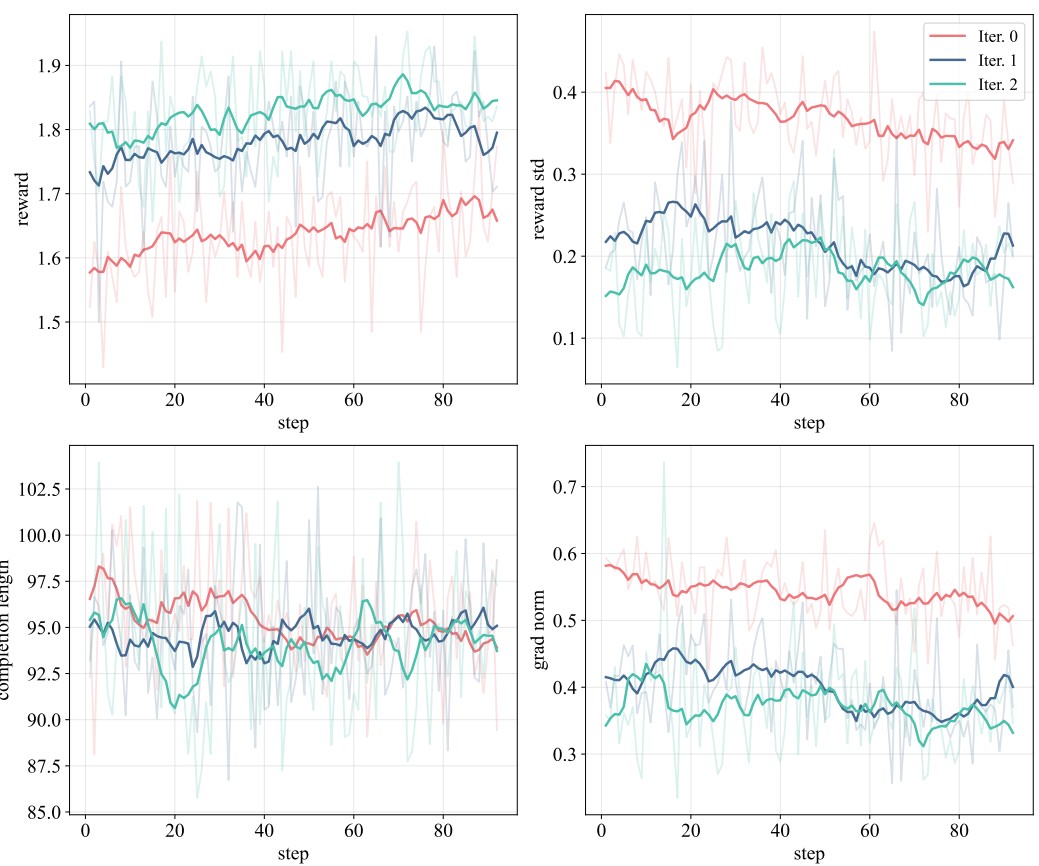

Figure 6: Visualization of the training process over various iterations. We present the changes in key training metrics, including total reward, reward standard deviation, response length, and gradient norm with respect to training steps.

### C.1   The training dynamics of URST

To provide more details, we report key metrics from our training process across multiple iterations, as shown in Figure 6.

Across all three training iterations (Iteration 0, 1, and 2), total reward shows a gradual upward trend as training progresses. The improvement from Iteration 0 to Iteration 1 is particularly notable, while subsequent iterations display more modest and convergent changes. This pattern suggests that the model adapts and learns more effectively during the earlier stages of training, and as training progresses, the rate of improvement stabilizes.

Both the reward standard deviation and gradient norm exhibit a clear downward trend over training steps and across iterations. This suggests that the model becomes more stable and less volatile as

training proceeds. As the training continues, these metrics converge, reflecting increased confidence and consistency in the model's behavior.

Response length remains relatively stable across all iterations, without any significant increasing or decreasing trend. This is expected, as our setup employs a cold-start initialization. Consequently, the reasoning patterns, especially regarding output length and format, exhibit relatively limited variation during the training process.

## C.2 Statistical significance and stability

In the Table 10, we present the statistical results of SFT and URST with three different random seeds. URST significantly outperforms SFT across all three test sets. These improvements are statistically meaningful given the non-overlapping confidence intervals, particularly for the F1 score, where URST shows a substantial gain (e.g., +4.82 on AITW-OOD-traj). Moreover, the smaller standard deviations in most metrics indicate that URST produces more stable results across different runs. Overall, these findings demonstrate that URST not only improves performance but also enhances result consistency.

Table 10: The statistical results of SFT and URST with three different random seeds

| Methods | SFT | | URST | |
|---|---|---|---|---|
| | Acc. | F1 | Acc. | F1 |
| AITW-ID-traj | 83.61±0.79 | 87.15±0.50 | 87.78±0.39 | 90.64±0.27 |
| AITW-OOD-traj | 93.61±1.42 | 78.71±3.00 | 94.72±0.78 | 83.53±2.09 |
| AW-OOD-traj | 77.13±1.17 | 71.30±1.17 | 83.40±1.27 | 82.14±0.81 |
| Overall | 83.08±0.71 | 78.62±0.71 | 87.47±0.81 | 85.59±1.05 |

## C.3 The evaluation of evaluator

As shown in Figure 3, we present the overall performance across multiple iterations. It can be observed that all iterations consistently enhance the evaluator's performance. However, performing additional iterations yields diminishing gains due to the relatively low quality of the newly sampled training data. In the Table 11, we also provide the detailed results of multiple iterations. The performance of URST on AITW-ID-traj and AW-OOD-traj gradually increases with iterative training. However, its performance on AITW-OOD-traj peaks at iteration 0 and exhibits fluctuations (primarily in the F1 score) in subsequent iterations. We attribute this behavior to the severe class imbalance in AITW-OOD-traj (success: 14%, fail: 86%), as shown in Table 7. The F1 score is highly sensitive to even small performance variations in the minority class.

Table 11: The detailed results of URST across different iterations.

| Phase | AITW-ID-traj | | AITW-OOD-traj | | AW-OOD-traj | | Overall | |
|---|---|---|---|---|---|---|---|---|
| | Acc. | F1 | Acc. | F1 | Acc. | F1 | Acc. | F1 |
| Initial | 80.00 | 82.85 | 95.00 | 81.25 | 79.82 | 73.05 | 83.80 | 77.88 |
| Iter. 0 | 83.33 | 86.11 | 96.67 | 87.50 | 78.48 | 72.41 | 84.45 | 79.43 |
| Iter. 1 | 86.67 | 89.47 | 93.33 | 78.95 | 81.61 | 78.31 | 85.95 | 82.85 |
| Iter. 2 | 87.50 | 90.45 | 94.16 | 82.05 | 81.62 | 79.60 | 86.39 | 84.13 |

## C.4 Generalization beyond mobile GUI tasks

Recently proposed AgentRewardBench [25] is the first benchmark to assess the effectiveness of MLLM-based evaluators for evaluating web agents. It contains 1302 web trajectories across 5 benchmarks and 4 agents. Due to the lack of training data, we split AgentRewardBench into two sets, 1047 trajectories for the training set and 255 for the test set, and ensure a similar class distribution, as shown in Table 12.

We evaluate the prompting-based (WebJudge, AutoEval*) and fine-tuning-based (SFT, URST) methods on this web benchmark, and report the results in the Table 13. URST is trained under the

Table 12: The statistics of trajectory evaluation datasets constructed from AgentReward-Bench [25].

| Datasets | #Success | #Total | Success Rate |
|---|---|---|---|
| Training | 1047 | 293 | 27.99% |
| Test | 255 | 63 | 24.71% |

Table 13: Comparison of our URST with prompting-based and fine-tuned evaluators on AgentRewardBench [25].

| Methods | AgentRewardBench | |
|---|---|---|
| | Acc. | F1 |
| WebJudge | 83.07 | 68.15 |
| AutoEval* | 85.10 | 74.67 |
| SFT | 84.31 | 74.36 |
| URST | 87.06 | 77.85 |

setting of initial-300 samples, iter. 0-200 samples, iter. 1-200samples, and SFT is trained with the same total number of samples. Our URST achieves the highest performance on AgentRewardBench, reaching 87.06% accuracy and 77.85% F1 score, substantially outperforming the other baselines. These comparisons underscore the effectiveness and generalization of URST.

## C.5 Case study

Here we present more qualitative cases comparing URST with the SFT baseline in trajectory evaluation tasks, as shown in Figure 7.

- **Example 1 (left).** The SFT model incorrectly concludes that the task is completed, reasoning that the appearance of a protection plan pop-up "typically appears after adding an item to the cart." However, this might be a case of over-reliance on common patterns without a comprehensive check. In contrast, URST correctly identifies that the product has not actually been added to the cart, noting that the last screen only shows the protection plan dialog "instead of the actual product being added to the cart." This shows URST's ability to discern subtle differences in interface feedback.

- **Example 2 (middle).** The SFT model falsely claims that "the option to display folders first is not selected," resulting in a wrong NO judgment. While URST provides a more accurate analysis. It grounds its reasoning directly in the UI evidence, avoiding unsupported assumptions. This case highlights URST's advantage in mitigating hallucinations.

- **Example 2 (right).** The SFT model incorrectly concludes that the task is incomplete due to the "lack of explicit confirmation" in the last two screen states. This reflects a weakness in tracking fine-grained transitions and recognizing subtle but semantically meaningful changes in the GUI elements. In contrast, URST successfully identifies this transition by integrating an action history that includes the precise click location, demonstrating a strong sensitivity to GUI elements and their associated semantic cues.

To further analyze the limitations of our proposed method, we present several representative failure cases observed during trajectory evaluation, as illustrated in Figure 8.

- **Example 1 (left): Failure Due to Insufficient Observation.** The evaluator is unable to determine whether the correct item has been deleted, as the last two visual states and action history do not provide sufficient evidence to verify the deletion of the specific entry "New Jacket". While a deletion action is present, there is no direct indication that it affected the correct item in the last two states. This case exposes the trade-off between limited observation and reasoning accuracy: relying on only the last two states can reduce computational overhead, but sometimes omits essential information for task validation. Finding a balance between context efficiency and task-specific observability remains an open challenge.

- **Example 2 (middle): Failure Due to GUI elements Misinterpretation.** Although the evaluator correctly identifies the selection of the paint bucket tool and color, it falsely concludes the task is complete. The confusion arises from an orange cursor widget visible at the bottom of the screen, which the model incorrectly interprets as the filled canvas area. This case highlights a common failure mode where the model lacks domain-specific visual understanding, confusing GUI indicators (like cursors or tool previews) with actual content

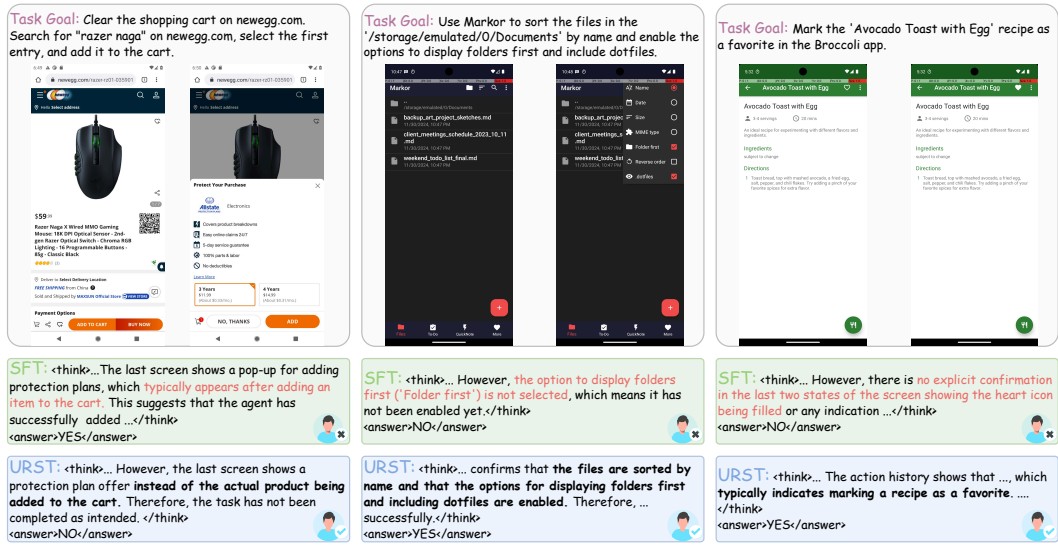

Figure 7: Illustration of some trajectory evaluation tasks with the model-generated reasoning. URST produces correct judgments through precise reasoning, whereas SFT fails.

(like painted canvas). This failure case underscores the importance of equipping the model with more comprehensive domain-specific knowledge about apps and a better understanding of GUI elements.

- **Example 3 (right): Failure Due to Instruction Misunderstanding.** While the evaluator correctly identifies that the default search engine has been changed to "Yahoo!", it overlooks the second part of the instruction, "and then return to the main settings menu." The final screen still shows the search engine selection page, indicating that the task is only partially complete. This failure case suggests that the evaluator sometimes lacks robust multi-step instruction comprehension. It underscores the need for improved task decomposition and intent modeling in trajectory evaluation settings.

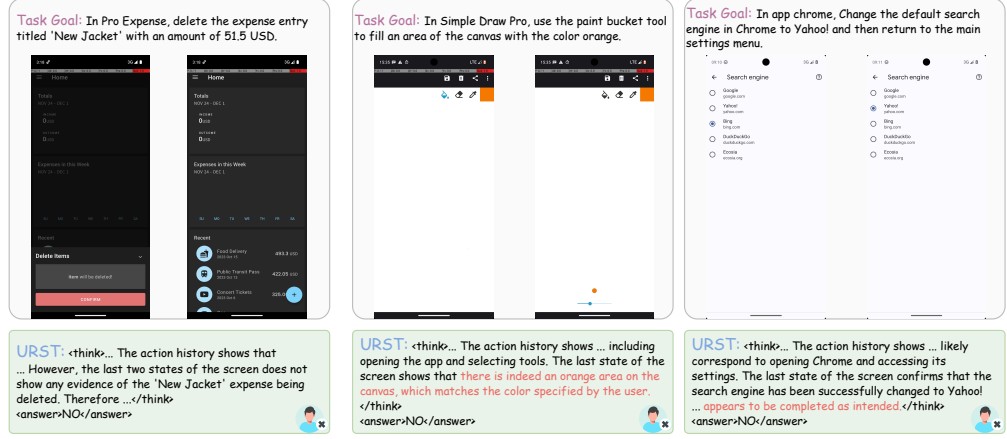

Figure 8: Illustration of failure cases observed in trajectory evaluation.

## D  Limitations and Potential Social Impacts

There are multiple potential limitations in this work. First, we only fine-tuned one family of models, Qwen2-VL / Qwen2.5-VL; however, while the absolute performance values would change, we expect our relative findings to be consistent across model families. Additionally, while our results suggest

that CoT reasoning is crucial to the performance improvement of evaluator training with RL, we have not extensively explored the effects of different prompting techniques in this work, which will be an interesting future direction. Further, due to computational limitations, our evaluator is trained only with tasks from AITW instead of all possible tasks on the mobile or computer device. Our design of the URST algorithm aims for maximal implementation efficiency and effectiveness, so we hope that our approach to serve as a base algorithm for future research to build on, including algorithmic research as well as expanding the space of tasks.

The URST framework enhances data quality through rigorous trajectory filtering, leading to agents trained on cleaner, higher-quality examples, thus minimizing erroneous or harmful behaviors during deployment. The framework relies on lightweight MLLMs and avoids dependence on proprietary models, potentially democratizing access to powerful trajectory evaluation tools for institutions or developers with limited resources. However, we also acknowledge that this work has some potential negative impacts. If the initial training data contains biases, the self-improvement loop could reinforce these issues, potentially resulting in unfair or incorrect judgments, especially in edge cases or unfamiliar interface designs. Advanced GUI agents trained with URST could potentially be used to manipulate user interfaces in unethical ways if deployed without proper safeguards.

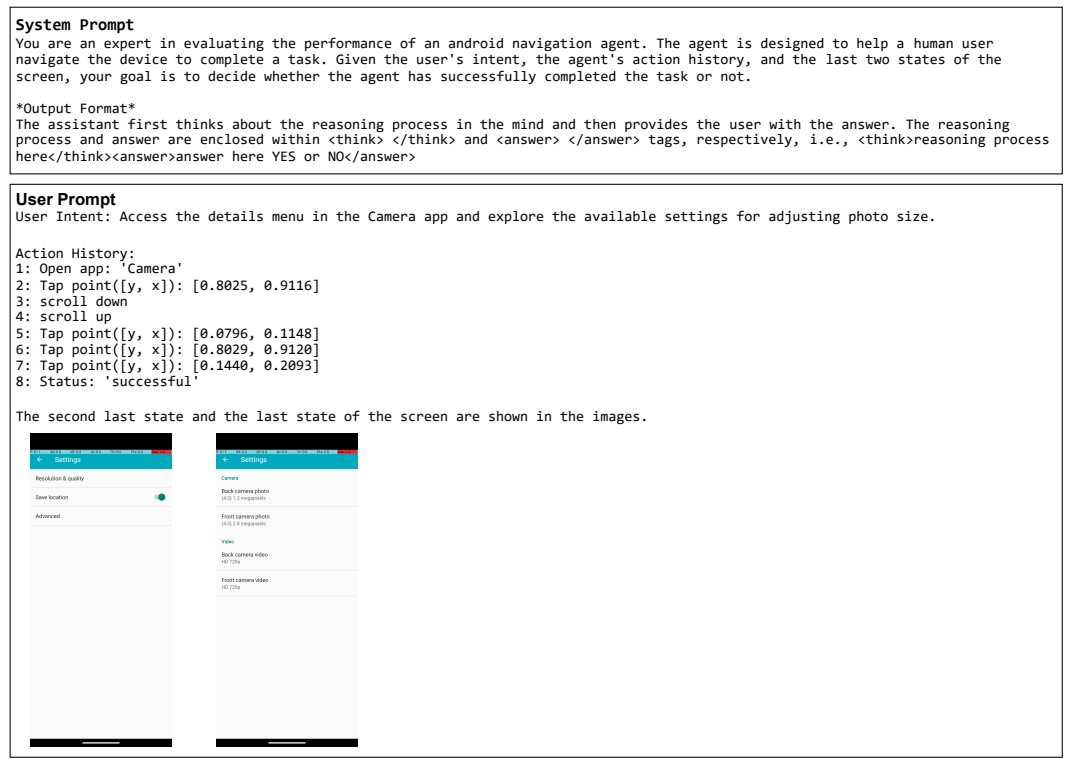

Figure 9: Prompt example for fine-tuning-based methods.

Figure 10: Prompt template for AgentTrek.

1. Task Instruction: This is a clear and specific directive provided in natural language, detailing the online activity to be carried out.

2. Result Screenshots: This is a visual representation of the screen showing the result or intermediate state of performing a task. It serves as visual proof of the actions taken in response to the instruction.

3. Result Response: This is a textual response obtained after the execution of the task. It serves as textual result in response to the instruction.

-- You DO NOT NEED to interact with screens or perform actions.
-- You SHOULD NOT make assumptions based on information not presented in the screenshot when comparing it to the instructions.
-- Your primary responsibility is to conduct a thorough assessment of the task instruction against the outcome depicted in the screenshot and in the response, evaluating whether the actions taken align with the given instructions.
-- NOTE that the instruction may involve more than one task, for example, locating the garage and summarizing the review. Failing to complete either task, such as not providing a summary, should be considered unsuccessful.
-- NOTE that the screenshot is authentic, but the response provided by LLM is generated at the end of state, and there may be discrepancies between the text and the screenshots.
-- Note the difference: 1) Result response may contradict the screenshot, then the content of the screenshot prevails, 2) The content in the Result response is not mentioned on the screenshot, choose to believe the content.

You should elaborate on how you arrived at your final evaluation and then provide a definitive verdict on whether the task has been successfully accomplished, either as 'SUCCESS' or 'FAILURE'.

**User Prompt**
TASK: {task_goal}

Result Response: {last_action}

{num_of_all_states} screenshots at the end:

Figure 11: Prompt template for WebVoyager.

```
┌─────────────────────────────────────────────────────────────────────────────────────────┐
│ Key Point Identification Prompt                                                           │
│ You are an expert tasked with analyzing a given task to identify the key points explicitly stated in the task description. │
│                                                                                           │
│ **Objective**: Carefully analyze the task description and extract the critical elements explicitly mentioned in the task for │
│ achieving its goal.                                                                       │
│                                                                                           │
│ **Instructions**:                                                                         │
│ 1. Read the task description carefully.                                                   │
│ 2. Identify and extract **key points** directly stated in the task description.           │
│   - A **key point** is a critical element, condition, or step explicitly mentioned in the task description. │
│   - Do not infer or add any unstated elements.                                            │
│   - Words such as "best," "highest," "cheapest," "latest," "most recent," "lowest," "closest," "highest-rated," "largest," and │
│ "newest" must go through the sort                                                         │
│ function(e.g., the key point should be "Filter by highest").                              │
│                                                                                           │
│ **Respond with**:                                                                         │
│ - **Key Points**: A numbered list of the explicit key points for completing this task, one per line, without explanations or │
│ additional details.                                                                       │
│                                                                                           │
│ Task: {task_goal}                                                                         │
└─────────────────────────────────────────────────────────────────────────────────────────┘

┌─────────────────────────────────────────────────────────────────────────────────────────┐
│ Key Screenshot Identification Prompt                                                      │
│ You are an expert evaluator tasked with determining whether an image contains information about the necessary steps to complete │
│ a task.                                                                                   │
│                                                                                           │
│ **Objective**: Analyze the provided image and decide if it shows essential steps or evidence required for completing the task. │
│ Use your reasoning to explain your decision before assigning a score.                     │
│                                                                                           │
│ **Instructions**:                                                                         │
│ 1. Provide a detailed description of the image, including its contents, visible elements, text (if any), and any notable features. │
│                                                                                           │
│ 2. Carefully examine the image and evaluate whether it contains necessary steps or evidence crucial to task completion: │
│ - Identify key points that could be relevant to task completion, such as actions, progress indicators, tool usage, applied │
│ filters, or step-by-step instructions.                                                    │
│ - Does the image show actions, progress indicators, or critical information directly related to completing the task? │
│ - Is this information indispensable for understanding or ensuring task success?           │
│ - If the image contains partial but relevant information, consider its usefulness rather than dismissing it outright. │
│                                                                                           │
│ 3. Provide your response in the following format:                                         │
│ - **Reasoning**: Explain your thought process and observations. Mention specific elements in the image that indicate necessary │
│ steps, evidence, or lack thereof.                                                         │
│ - **Score**: Assign a score based on the reasoning, using the following scale:            │
│    - **1**: The image does not contain any necessary steps or relevant information.       │
│    - **2**: The image contains minimal or ambiguous information, unlikely to be essential. │
│    - **3**: The image includes some relevant steps or hints but lacks clarity or completeness. │
│    - **4**: The image contains important steps or evidence that are highly relevant but not fully comprehensive. │
│    - **5**: The image clearly displays necessary steps or evidence crucial for completing the task. │
│                                                                                           │
│ Respond with:                                                                             │
│ 1. **Reasoning**: [Your explanation]                                                      │
│ 2. **Score**: [1-5]                                                                        │
│                                                                                           │
│ Task**: {task_goal}                                                                       │
│                                                                                           │
│ **Key Points for Task Completion**: {key_points}                                          │
│                                                                                           │
│ The snapshot of the screen is shown in the image.                                         │
└─────────────────────────────────────────────────────────────────────────────────────────┘

┌─────────────────────────────────────────────────────────────────────────────────────────┐
│ Outcome Judgement Prompt                                                                   │
│ You are an expert in evaluating the performance of an android navigation agent. The agent is designed to help a human user │
│ navigate the device to complete a task. Given the user's task, the agent's action history, key points for task completion, some │
│ potentially important states in the agent's trajectory and their reasons, your goal is to determine whether the agent has │
│ completed the task and achieved all requirements.                                         │
│                                                                                           │
│ Your response must strictly follow the following evaluation criteria!                     │
│ *Important Evaluation Criteria*:                                                          │
│ 1: The filtered results must be displayed correctly. If filters were not properly applied (i.e., missing selection, missing │
│ confirmation, or no visible effect in results), the task is not considered successful.    │
│ 2: You must carefully check whether these snapshots and action history meet these key points. Ensure that specific filter │
│ conditions, such as "best," "highest," "cheapest," "latest," "most recent," "lowest," "closest," "highest-rated," "largest," and │
│ "newest" are correctly applied using the filter function(e.g., sort function).            │
│ 3: Certain key points or requirements should be applied by the filter. Otherwise, a search with all requirements as input will be │
│ deemed a failure since it cannot guarantee that all results meet the requirements!        │
│ 4: If the task requires filtering by a specific range of money, years, or the number of beds and bathrooms, the applied filter │
│ must exactly match the given requirement. Any deviation results in failure. To ensure the task is successful, the applied filter │
│ must precisely match the specified range without being too broad or too narrow.           │
│ Examples of Failure Cases:                                                                │
│ - If the requirement is less than $50, but the applied filter is less than $25, it is a failure. │
│ - If the requirement is $1500-$2500, but the applied filter is $2000-$2500, it is a failure. │
│ - If the requirement is $25-$200, but the applied filter is $0-$200, it is a failure.     │
│ - If the required years are 2004-2012, but the filter applied is 2001-2012, it is a failure. │
│ - If the required years are before 2015, but the applied filter is 2000-2014, it is a failure. │
│ - If the task requires exactly 2 beds, but the filter applied is 2+ beds, it is a failure. │
│ 5: Some tasks require a submission action or a display of results to be considered successful. │
│ 6: If the retrieved information is invalid or empty(e.g., No match was found), but the agent has correctly performed the required │
│ action, it should still be considered successful.                                         │
│ 7: If the current page already displays all available items, then applying a filter is not necessary. As long as the agent selects │
│ items that meet the requirements (e.g., the cheapest or lowest price), the task is still considered successful. │
│                                                                                           │
│ *IMPORTANT*                                                                               │
│ Format your response into two lines as shown below:                                       │
│                                                                                           │
│ Thoughts: <your thoughts and reasoning process based on double-checking each key points and the evaluation criteria> │
│ Status: "success" or "failure"                                                            │
│                                                                                           │
│ User Task: {task_goal}                                                                    │
│                                                                                           │
│ Key Points: {key_points}                                                                  │
│                                                                                           │
│ Action History: {history_actions}                                                         │
│                                                                                           │
│ The potentially important snapshots of the screen in the agent's trajectory and their reasons: {thoughts} │
└─────────────────────────────────────────────────────────────────────────────────────────┘
```

Figure 12: Prompt template for WebJudge.

**System Prompt**
You are an expert in evaluating the performance of an android navigation agent. The agent is designed to help a human user navigate the device to complete a task. Given the user's intent, and the final state of the screen, your goal is to decide whether the agent has successfully completed the task or not.

*IMPORTANT*
Format your response into two lines as shown below:

Thoughts: <your thoughts and reasoning process>"
Status: "success" or "failure"

**User Prompt**
User Intent: {task_goal}

Action History: {history_actions}

The last snapshot of the screen is shown in the image.

Figure 13: Prompt template for AutoEval.

**System Prompt**
You are an expert in evaluating the performance of an android navigation agent. The agent is designed to help a human user navigate the device to complete a task. Given the user's intent, the agent's action history, and the last two states of the screen, your goal is to decide whether the agent has successfully completed the task or not.

*Output Format*
Format your response into two lines as shown below:

Thoughts: <your thoughts and reasoning process>
Status: <YES or NO>

**User Prompt**
User Intent: {task_goal}

Action History: {history_actions}

The second last state and the last state of the screen are shown in the images.

Figure 14: Prompt template for AutoEval*, which is constructed by enhancing AutoEval with the last two screenshots.

**System Prompt**
You are a smart and helpful visual assistant that is well trained to manipulate mobile phones.
Your task is to navigate on the current screen to complete the user request.
- You are provided with a screenshot of the current mobile phone.
- You are provided with history actions trying to accomplish the user request.
- You are required to decide on the next single-step valid action to conduct on the current screen.

## Valid action types on the screen
- DUAL_POINT
- TYPE
- PRESS_BACK
- PRESS_HOME
- PRESS_ENTER
- STATUS_TASK_COMPLETE

## Output Format
Your response should be strictly structured in JSON format, consisting of the following keys and corresponding content:
{
    "action_type": "<string, your action decision action type>",
    "touch_point": "<list, coordinates for the touch point on the screen, [y, x], when action_type is not DUAL_POINT, it should be [-1, -1]>",
    "lift_point": "<list, coordinates for the lift point on the screen, [y, x], when action_type is not DUAL_POINT, it should be [-1, -1]>",
    "type_text": "<string, the text to type on the screen, only for TYPE action type, otherwise it should be an empty string>"
}

## Output Example1
{"action_type": "DUAL_POINT", "touch_point": [0.8949, 0.2941], "lift_point": [0.8949, 0.2941], "type_text": ""}

## Output Example2
{"action_type": "DUAL_POINT", "touch_point": [0.8, 0.5], "lift_point": [0.2, 0.5], "type_text": ""}

## Output Example3
{"action_type": "TYPE", "touch_point": [-1.0, -1.0], "lift_point": [-1.0, -1.0], "type_text": "LiveIn"}

## Output Example4
{"action_type": "PRESS_BACK", "touch_point": [-1.0, -1.0], "lift_point": [-1.0, -1.0], "type_text": ""}

## Output Example5
{"action_type": "PRESS_HOME", "touch_point": [-1.0, -1.0], "lift_point": [-1.0, -1.0], "type_text": ""}

## Output Example6
{"action_type": "PRESS_ENTER", "touch_point": [-1.0, -1.0], "lift_point": [-1.0, -1.0], "type_text": ""}

## Output Example7
{"action_type": "STATUS_TASK_COMPLETE", "touch_point": [-1.0, -1.0], "lift_point": [-1.0, -1.0], "type_text": ""}

**User Prompt**
<image>
## Actions History
{history_actions}
## Generate next actions to do this task.

Figure 15: Prompt template for GUI navigation agent training with AITW dataset.

**User Prompt**

<image>
You are a GUI task expert, I will provide you with a high-level instruction, an action history, a screenshot with its corresponding accessibility tree.

High-level instruction: {high_level_instruction}

Action history: {action_history}

Accessibility tree: {accessibility_tree}

Please generate the low-level thought and action for the next step.

Figure 16: Prompt template for GUI navigation agent training with OS-Genesis dataset.

