# OpenReview forum: "Enhancing GUI Agent with Uncertainty-Aware Self-Trained Evaluator"
_NeurIPS.cc/2025/Conference — NeurIPS 2025 poster_

### Official Review · Reviewer_nyzi · 2025-06-13

**Clarity:** 3
**Significance:** 2
**Originality:** 3
**Rating:** 4
**Confidence:** 4

**Summary:**

This paper presents a framework, Uncertainty-aware Reinforced Self-Training, for training lightweight Multimodal Large Language Models to serve as efficient and reliable evaluators for GUI agent navigation trajectories. Addressing the limitations of costly and domain-agnostic prompting-based methods that rely on proprietary MLLMs, URST enables an evaluator to improve itself iteratively by learning from its own generated thoughts and judgments. The framework introduces two key technical contributions: a novel uncertainty-aware sampling strategy that uses prediction entropy to select the most informative training examples, and a Simplified Group Policy Optimization algorithm that effectively leverages both positive and negative samples to enhance the model's reasoning. The authors demonstrate that their URST-trained evaluator not only surpasses the performance of prompting-based and other fine-tuning methods on both in-domain and out-of-domain datasets but also leads to consistent performance improvements in downstream GUI agents when used as a data filter.

**Questions:**

- Line 123: why is this approach/formulation restricted to “offline” dataset online?

- Equation 1: why does the evaluator need to take in the chain-of-thought c as input?

- Can you elaborate on how the “Prompting-based Methods” are implemented? For example, for AgentTrek, did you just use gpt-4o-mini and fasttext classifier to filter trajectories and then train the model on the filtered dataset?

**Ethical Concerns:**

["NO or VERY MINOR ethics concerns only"]

**Final Justification:**

The reviewer has addressed all my concerns and thus I have increased my ratings.

**Limitations:**

yes

**Quality:**

2

**Strengths And Weaknesses:**

__Strengths__:
- The writing of the paper is clear, making it easy to follow.

- The problem formulation is interesting and novel.


__Weaknesses__:

- The biggest concern of this work is insufficient baseline comparisons. In Table 1, the paper should also report the performance of non-finetuned models, (namely, Qwen2VL-2B, Qwen-VL-Max, gpt-4o etc) for better understand the effectiveness of their proposed training methods and advantages over proprietary models. Additionally, since the proposed approach also performs some forms of distillation of larger models (line 167), the author should also compare with other simpler distillation-based methods so that readers can better appreciate the proposed method.


- Also critically, performance enhancement is quite minimal in both in-domain and cross-domain when it comes to actually enhancing GUI agent performance. In Section 4.4, the improvement is < 2% on in-domain and <1% on out-of-domain. Based on the results, it is hard to convince that the trajectory evaluator is a must/effective in enhancing GUI agent performance.

- For section 4.4, the author should also compare their approach with others. In fact, the comparison with baselines here is more important than Table 1 as they reflect how different approaches are able to improve GUI agent performance, which is the ultimate goal of building a trajectory evaluator.

---

> ### Author Rebuttal · Authors · 2025-07-31
>
> We appreciate the reviewer’s insightful comments and helpful suggestions. Below are our responses and explanations:
>
> ---
>
> > **Q1.** The biggest concern of this work is insufficient baseline comparisons.
>
> **A1.** Thanks for your comments. We would like to clarify this point about insufficient baseline comparisons. **The prompting-based methods (AgentTrek, WebVoyager, WebJudge, and AutoEval) are non-finetuned models** because they use proprietary MLLMs that cannot be trained (as described in Lines 131-133). These prompting-based methods are based on Qwen-VL-Max, and show inferior performances to our URST, demonstrating the effectiveness of URST.
>
> **“SFT” adopted in our work is a typical distillation-based method** (as described in Lines 248-249). It uses the predictions (CoTs and judgments) from a larger model, Qwen-VL-Max to train Qwen2.5VL-3B for knowledge distillation. “SFT” also achieves inferior results, highlighting the value of our URST. For clarification, we will revise our manuscript and add more detailed descriptions.
>
> ---
>
> > **Q2&3.** It is hard to convince that the trajectory evaluator is a must/effective in enhancing GUI agent performance. For section 4.4, the author should also compare their approach with others. In fact, the comparison with baselines here reflects how different approaches are able to improve GUI agent performance, which is the ultimate goal of building a trajectory evaluator.
>
> **A2&3.** We thank the reviewer for raising this important point about the goal of building a trajectory evaluator. Drawing on prior studies [1-3] and our practical insights, we identify **three key roles of the trajectory evaluator for enhancing GUI agents**.
>
> ### **1. An autonomous evaluator for online environments**
>
> It eliminates the need for manually written verification scripts and has greater scalability and adaptability. The primary contribution of our work is to build this kind of evaluator for assessing the agent-executed trajectories (such as samples in AITW-OOD-traj and AW-OOD-traj), showing the effectiveness and superiority of our URST.
>
> ### **2. A filter for cleaning the noisy dataset**
>
> By filtering the incorrect trajectories, we can improve the quality and reliability of the dataset. The impact of the filter increases as the noise level in the data rises. To prove this point, we conducted some comparison experiments by leveraging two settings, Standard-Noise and High-Noise.
>
> - **Standard-Noise**: Discarding all incorrect trajectories, and using all steps in each trajectory for training GUI agents.
>
> - **High-Noise**: Discarding all incorrect trajectories, and using the last 40% of steps in each trajectory for training GUI agents. **The underlying rationale is that the majority of incorrect trajectories tend to fail in the last few steps.** When tasks are highly challenging or involve unseen environments, the data automatically collected by the agent often contains high levels of noise, meaning that the entire trajectory is wrong from the very beginning.
>
> Based on the following Table, we can conclude that 1) our URST can consistently outperform other baselines (including AutoEval* and SFT) in filtering noisy datasets. 2) When the noisy level rises, all filters (based on AutoEval*, SFT, and URST) can achieve a greater impact on agent performance, with URST consistently achieving the best result by a significant margin. 3) URST demonstrates a clear overall advantage, delivering optimal outcomes in time cost, expenditure, and performance.
>
> | Methods | Time Cost (s / per sample) | Cost ($) (per 1M Tokens) | AITW |  | OS-Genesis |  |
> | --- | --- | --- | --- | --- | --- | --- |
> |  |  |  | Standard-Noise | **High-Noise** | Standard-Noise | **High-Noise** |
> | Original | 0 | 0 | 63.50 | 20.88 | 46.83 | 46.43 |
> | Filtered by AutoEval* | 5.99 | 0.41 | 64.98 | 38.93 | 47.26 | 47.65 |
> | Filtered by SFT | 2.17 | 0 | 64.32 | 32.59 | 47.18 | 47.22 |
> | Filtered by URST | 2.17 | 0 | 65.02 | 40.41 | 47.66 | 48.26 |
>
> ### **3. A reward signal for agent inference enhancement**
>
> In an online environment, the trajectory evaluator can serve as a reward signal to enhance an existing GUI agent at inference time, using the Reflexion technique [4]. Specifically, a GUI agent first attempts a task, and an external evaluator is used to judge whether its attempt was successful or not. The agent will be prompted to reflect on the failure and retry unsuccessful tasks. We conduct experiments on the most popular online benchmark, AndroidWorld, with M3A agents. The best prompting-based method, AutoEval*, and our URST are used as a reward signal for eliciting reflection. URST achieves superior performance, with a 1.8% improvement over AutoEval*, underscoring its clear advantage in enhancing GUI agents.
>
> |  | AndoridWorld successful rate |
> | --- | --- |
> | M3A | 39.7% |
> | M3A + AutoEval* | 44.8% |
> | **M3A + URST** | **46.6%** |
>
> [1] Autonomous Evaluation and Refinement of Digital Agents, COLM 2024.
>
> [2] AgentTrek: Agent Trajectory Synthesis via Guiding Replay with Web Tutorials, ICLR 2025.
>
> [3] An Illusion of Progress? Assessing the Current State of Web Agents, COLM 2025.
>
> [4] Reflexion: language agents with verbal reinforcement learning, NeurIPS 2023.
>
> ---
>
> > **Q4.** Line 123: Why is this approach/formulation restricted to “offline” dataset online?
>
> **A4.** For training a trajectory evaluator, we need a collected dataset containing GUI navigation trajectories. This navigation dataset can be collected in an online environment. When using this collected dataset, we do not need online interaction, thus it can be treated as an “offline” dataset.
>
> ---
>
> > **Q5.** Equation 1: Why does the evaluator need to take in the chain-of-thought c as input?
>
> **A5.** Thanks for pointing it out. This is a typo, we will revise equation (1) by removing chain-of-thought c from the inputs.
>
> ---
>
> > **Q6.** Can you elaborate on how the “Prompting-based Methods” are implemented?
>
> **A6.** The implementation details of prompting-based methods are described in Figures 5-9 in the Appendix. We also provide the codes for reproduction. For AgentTrek, we follow the same setting in the original paper for evaluating the GUI navigation trajectory. The GPT-4o-mini and fasttext classifiers adopted in AgentTrek are used to filter textual tutorials (for better data collection), instead of GUI navigation trajectories. Thus, we exclude these steps.

---

> ### Comment · Reviewer_nyzi · 2025-08-03
>
> Thank you for the detailed response, which addressed most of my comments and I will raise my score to 4. I have some follow-up questions:
>
> - Q1: since the prompting-based baselines use the same base models, do they differ in the prompts they use? or are there other differences?
> - Q2: how did you construct the high-noise instances?

---

> > ### Author Response · Authors · 2025-08-05
> >
> > Thanks for your feedback and for raising your score. We truly appreciate it!
> >
> > To answer your follow-up questions:
> >
> > **A1.** Yes, the prompting-based baselines differ in their prompts, screenshot inputs, and workflows. Specifically,
> >
> > - WebVoyager takes all screenshots in a trajectory as input.
> > - AutoEval and AgentTrek focus on the final screenshot.
> > - AutoEval* extends its input scope to include the last two screenshots.
> > - WebJudge employs a key screenshot identification strategy (based on a proprietary MLLM) to extract the most critical screenshots.
> >
> > Due to these differences in input selection, each method also uses distinct prompts and workflows.
> >
> > **A2.** The construction of high-noise datasets stems from our observations and practical insights. We found that the majority of incorrect trajectories tend to fail in the last few steps. In other words, the latter part of the trajectory contains more noise. Therefore, we conduct the filtering on the whole GUI trajectory, but only use the last 40% of steps in each trajectory as the training data for GUI agents. Compared to the original dataset containing the last 40% of steps, our filtered datasets exhibit significantly improved trajectory quality. This high-noise phenomenon similarly manifests in datasets gathered from highly challenging or unseen environments, where incorrect trajectories often fail early in execution.

---

### Official Review · Reviewer_oZh1 · 2025-06-21

**Clarity:** 3
**Significance:** 2
**Originality:** 3
**Rating:** 4
**Confidence:** 4

**Summary:**

This paper proposes a framework for developing evaluators for GUI agents. Unlike API--based evaluators, which are expensive or small LM based evaluators which may not be as good, it seeks to develop better SLM-based judges. The key innovation is an iterative self-training based approach to select quality examples for training MLLMs. It uses uncertaining-aware reinforced training to select quality examples that add more value. The results are reported on AITW-ID, AITW-OOD, and a benchmark based on AndroidWorld created by authors.

**Questions:**

1. Is this approach applicable just to GUI agents or LLM agents in general? I see no design considerations specific to GUI agents. It would be great if the authors can evaluate this on LLM agent benchmarks in general e.g. JudgeBench [1].

[1] Tan et al., JudgeBench: A Benchmark for Evaluating LLM-based Judges, ICLR 2025.

**Ethical Concerns:**

["NO or VERY MINOR ethics concerns only"]

**Final Justification:**

I read the author's response. However, I would like to see the results on the JudgeBench benchmark, which is very relevant for this problem. Hence, I would like to keep my score (borderline accept).

**Limitations:**

yes, limitations in appendix.

**Paper Formatting Concerns:**

Minor: missing period in Sec 2.3.
unnecessary capitalizations at some places.

**Quality:**

3

**Strengths And Weaknesses:**

Strengths:
1. Alleviates the issues of existing evaluators such as high cost of API-based evaluators
2. Makes use of negative samples as well in reinforced self-training
3. Improvement in GT agreement compared to existing baselines

Weaknesses:
1. Entropy is an indicator of confidence of evaluator. Although the entropy decreases (Fig. 4), it remains high.
2. The performance on standard benchmarks (AITW-ID, AITW-OOD) is 2-3%, which is marginal.

---

> ### Author Rebuttal · Authors · 2025-07-31
>
> We appreciate the reviewer’s insightful comments and helpful suggestions. Below are our responses and explanations:
>
> ---
>
> > **Q1.** Entropy is an indicator of confidence of evaluator. Although the entropy decreases (Fig. 4), it remains high.
>
> **A1.** Thanks for your insightful comments. As shown in Figure 4 of the main paper, the entropy of the training data sampled by URST is relatively high (a mean value of 0.4), which can be attributed to the goal of uncertainty-aware sampling: selecting uncertainty data with high entropy. If we measure the confidence of the evaluator, we should refer to the entropy of randomly sampled training data (a mean value of 0.02), which is very low and shows a strong confidence of the trained evaluator.
>
> ---
>
> > **Q2.** The performance on standard benchmarks (AITW-ID, AITW-OOD) is 2-3%, which is marginal.
>
> **A2.** Thanks for your insightful comments. In the following table, we present the results and inference time of prompting-based and fine-tuning-based methods. Compared to SFT, URST achieves significant performance improvements, especially on F1 scores (increases 3.49% - 4.82%). Compared to AutoEval*, URST obtains a slightly smaller performance improvement ( increases 2.64% - 3.53% on F1 score), but it delivers a 2.8x inference speedup. Overall, our URST achieves a well-balanced yet remarkable improvement in both performance and efficiency.
>
> | Methods | Inference Time (s / per sample) | AITW-ID-traj | AITW-OOD-traj |
> | --- | --- | --- | --- |
> | WebJudge | 38.1 | 85.91 | 66.67 |
> | AutoEval* | 5.99 | 88.00 | 80.00 |
> | SFT | 2.17 | 87.15 | 78.71 |
> | **URST** | **2.17** | **90.64** | **83.53** |
>
> ---
>
> > **Q3.** It would be great if the authors can evaluate this on LLM agent benchmarks in general e.g. JudgeBench.
>
> **A3.** We appreciate the reviewer’s insightful comment regarding the general applicability of our approach beyond GUI agents. While we fully agree that evaluating on broader LLM agent benchmarks such as JudgeBench would further strengthen our claims, conducting such extensive experiments is unfortunately beyond our current time and resource constraints. The evaluation setting and training data collection process of JudgeBench differ from those of our MobileGUIBench.
>
> Nevertheless, to address the reviewer’s concern about generalizability, we have extended URST to a **web-based benchmark** and conducted additional experiments. Recently proposed **AgentRewardBench** [1] is the first benchmark to **assess the effectiveness of MLLM-based evaluators for evaluating web agents**. It contains 1302 web trajectories across 5 benchmarks and 4 agents. Due to the lack of training data, we split AgentRewardBench into two sets, 1047 trajectories for the training set and 255 for the test set, and ensure a similar class distribution.
>
> We evaluate the prompting-based (WebJudge, AutoEval*) and fine-tuning-based (SFT, URST) methods on this web benchmark, and report the results in the following Table. URST is trained under the setting of initial-300 samples, iter. 0-200 samples, iter. 1-200samples, while SFT is trained with the same total number of samples. Our URST achieves the highest performance on AgentRewardBench, reaching 87.06% accuracy and 77.85 F1 score, substantially outperforming the other baselines. **The results demonstrate that URST maintains its effectiveness and exhibits strong generalization capabilities in a broader setting.**
>
> | Methods | AgentRewardBench |  |
> | --- | --- | --- |
> |  | Acc. | F1 |
> | WebJudge | 83.07 | 68.15 |
> | AutoEval* | 85.10 | 74.67 |
> | SFT | 84.31 | 74.36 |
> | **URST** | **87.06** | **77.85** |
>
> [1] AGENTREWARDBENCH: Evaluating Automatic Evaluations of Web Agent Trajectories, arxiv 2025.

---

> ### Author Response · Authors · 2025-08-05
>
> Dear Reviewer oZh1,
>
> Thank you again for the valuable comments. As the interactive discussion window will close soon, we kindly invite you to read our responses and let us know if you have any further questions.
> Thank you!
>
> Best regards,
>
> Authors

---

> > ### Comment · Reviewer_oZh1 · 2025-08-05
> > **Follow-up**
> >
> > Thanks a lot for your response. However, I would like to see the results on the JudgeBench benchmark, which is very relevant for this problem. Also, I am not sure why the numbers for A2 differ from those reported in Table 1.

---

> > > ### Author Response · Authors · 2025-08-06
> > >
> > > Thank you for your feedback. To answer your follow-up questions:
> > >
> > > **A1.** We sincerely appreciate the reviewer's valuable suggestion to evaluate our method on JudgeBench, which provides a framework for assessing LLM-based judges on response pairs across various domains. As the primary focus of our work is trajectory evaluation in GUI scenarios using MLLMs, JudgeBench initially are out of the scope. To further verify our URST, we are currently adapting our training and testing procedures to align with JudgeBench’s framework, and will provide these additional results as soon as possible.
> > >
> > > **A2.** In response to Q3 from Reviewer 3VNW, we conducted SFT and URST experiments with three different random seeds. The obtained mean values (90.64 on AITW-ID-traj and 83.53 on AITW-OOD-traj) are slightly higher than those originally reported in Table 1 of the main paper (90.45 and 82.05, respectively). We apologize for the confusion caused by our accidental misreporting of the mean values in the Rebuttal.

---

> > > > ### Author Response · Authors · 2025-08-09
> > > > **Evaluation on JudgeBench**
> > > >
> > > > Thank you for your valuable suggestions. We trained an LLM-based judge using the Skywork-Reward-Preference-80K dataset [1], and evaluated it on JudgeBench [2]. Due to resource and time constraints, we selected 1.1k samples from the original dataset for training, using Qwen2.5-3B as the base model. The table below presents the performance of zero-shot, distillation-based SFT and URST. We ensured that distillation-based SFT and URST were trained with the same amount of data. For distillation-based SFT, the training data is composed of reasoning outputs from GPT-4o. The results in the following Table show that our URST achieves superior performance through uncertainty-aware sampling and iterative reinforcement self-training.
> > > >
> > > > We follow the setting in the original paper [2] and evaluate the LLM-based judge twice, swapping the order of the response pairs in the second trial to mitigate the positional bias. A high inconsistency score suggests that the judge is either making guesses or lacks the ability to reliably distinguish between responses. We would be sincerely grateful if you would consider giving us a higher rating.
> > > >
> > > > | Methods | Acc. | Inconsistent |
> > > > | --- | --- | --- |
> > > > | Qwen2.5-3B | 22.57 | 67.14 |
> > > > | SFT | 33.71 | 46.29 |
> > > > | URST | 39.71 | 40.57 |
> > > >
> > > > [1] https://huggingface.co/datasets/Skywork/Skywork-Reward-Preference-80K-v0.2
> > > >
> > > > [2] JudgeBench: A Benchmark for Evaluating LLM-Based Judges, ICLR 2025

---

### Official Review · Reviewer_ktZw · 2025-07-02

**Clarity:** 2
**Significance:** 3
**Originality:** 2
**Rating:** 4
**Confidence:** 4

**Summary:**

This paper introduces URST, a framework for training MLLMs to evaluate GUI navigation trajectories more reliably. URST enables self-improvement through iterative self-generated judgments and uses an uncertainty-aware sampling strategy to select informative training examples. A simplified group policy optimization (SGPO) method helps the evaluator learn from diverse feedback. The resulting evaluator outperforms existing approaches on both in-domain and out-of-domain data and improves GUI agent training.

**Questions:**

**Generalization Beyond Mobile GUI Tasks**: The evaluation currently focuses only on mobile GUI tasks. To demonstrate broader applicability, it would be helpful to test the pipeline on other GUI domains, such as desktop or web interfaces. This would provide stronger evidence of the method's generalizability across varied interaction modalities.

**Ethical Concerns:**

["NO or VERY MINOR ethics concerns only"]

**Final Justification:**

The response resolves most of my concerns and I’m impressed by the consistent improvements across iterations.

**Limitations:**

See Weaknesses part.

**Quality:**

2

**Strengths And Weaknesses:**

## Strengths

The model achieves strong performance across multiple benchmarks. The writing is clear and the method is presented in a straightforward yet effective way. The motivation for self-improvement is particularly compelling in the GUI domain, where the cost of human annotation is high and scalable data generation is challenging.

## Weaknesses

**Reliance on Evaluator Uncertainty**: The method relies heavily on the uncertainty of the evaluator to guide trajectory selection. This implicitly assumes that the evaluator is sufficiently accurate, and when it expresses high confidence, the corresponding trajectory is likely of high quality. However, since the evaluator is not trained on any ground-truth data, its overall reliability is questionable. This casts doubt on whether its uncertainty is a trustworthy signal for trajectory assessment. A more thorough evaluation either automatic or human on how well the evaluator’s uncertainty correlates with actual trajectory quality would greatly strengthen the claim.

**Evaluator Evolution**: It would be valuable to track and report the performance trajectory of the evaluator itself over the course of training. Showing how its evaluation quality changes particularly in early iterations through either automatic metrics or human judgments would provide insights into the robustness and stability of the evaluator as it evolves.

**Alternative Sampling Metrics**: While evaluator uncertainty is used to select trajectories, it is not the only potential signal for quality. For instance, policy uncertainty could also serve as an indicator. If the policy model is uncertain about its decisions, the trajectory may be of lower quality. Comparing the effectiveness of evaluator uncertainty against alternative metrics (e.g., policy entropy) would offer a more comprehensive justification for the current choice and potentially reveal better alternatives.

**Baseline RL Methods**: Although the authors mention GRPO when introducing SGPO, there is no ablation or comparison with GRPO, PPO, or other standard reinforcement learning methods. This makes it unclear whether SGPO is tailored for the self-improvement framework or simply an arbitrary choice. Including these comparisons would help clarify the advantages of SGPO and whether it meaningfully outperforms existing baselines.

---

> ### Author Rebuttal · Authors · 2025-07-31
>
> We appreciate the reviewer’s insightful comments and helpful suggestions. Below are our responses and explanations:
>
> ---
>
> > **Q1.** **Reliance on Evaluator Uncertainty**: The method relies heavily on the uncertainty of the evaluator to guide trajectory selection. This casts doubt on whether its uncertainty is a trustworthy signal for trajectory assessment.
>
> **A1.** Thanks for your insightful comments. We would like to provide a detailed clarification. The uncertainty measurement of the evaluator aims to find the most valuable trajectories for the model, as trajectories with high uncertainty values reflect the limitations or knowledge gaps of the evaluator. The measurement of uncertainty does not involve the trajectory annotations and is irrelevant to the trajectory annotation quality.
>
> Although the evaluator is not trained on any ground-truth data, the initialization phase generates a strong base evaluator by distilling from a proprietary MLLM (generally has around an 80% F1 score). In subsequent iterations, the evaluator can leverage a reward model for judgment assessment, and gradually achieve better performance. As shown in the table of A3, we present a new metric for measuring uncertainty and also achieve a superior performance, demonstrating the effectiveness of our uncertainty-aware sampling.
>
> In summary, **the uncertainty is a useful signal for trajectory value assessment, and irrelevant to the trajectory annotation quality. The evaluator can achieve self-improvement with the help of distillation from a proprietary MLLM and guidance from the reward model despite the lack of ground-truth data.**
>
> ---
>
> > **Q2.** **Evaluator Evolution**: It would be valuable to track and report the performance trajectory of the evaluator itself over the course of training.
>
> **A2.** Thanks for your constructive suggestions. As shown in Figure 3 of the main paper, we present the overall performance across multiple iterations. **It can be observed that all iterations consistently enhance the evaluator’s performance. However, performing additional iterations yields diminishing gains due to the relatively low quality of the newly sampled training data.** In the following table, we also provide the detailed results of multiple iterations. The performance of URST on AITW-ID-traj and AW-OOD-traj gradually increases with iterative training. However, its performance on AITW-OOD-traj peaks at iteration 0 and exhibits fluctuations (primarily in the F1 score) in subsequent iterations. We attribute this behavior to the severe class imbalance in AITW-OOD-traj (success: 14%, fail: 86%), as shown in Table 1 of the Appendix. The F1 score is highly sensitive to even small performance variations in the minority class. We will include these detailed analyses in the revised version.
>
> |  | Total |  | AITW-ID-traj | (success: 64%, fail: 36%) | AITW-OOD-traj | (success: 14%, fail: 86%) | AW-OOD-traj | (success: 43%, fail: 57%) |
> | --- | --- | --- | --- | --- | --- | --- | --- | --- |
> |  | Acc | F1 | Acc | F1 | Acc | F1 | Acc | F1 |
> | Initial | 83.80 | 77.88 | 80.00 | 82.85 | 95.00 | 81.25 | 79.82 | 73.05 |
> | Iter. 0 | 84.45 | 79.43 | 83.33 | 86.11 | 96.67 | 87.50 | 78.48 | 72.41 |
> | Iter. 1 | 85.95 | 82.85 | 86.67 | 89.47 | 93.33 | 78.95 | 81.61 | 78.31 |
> | Iter. 2 | 86.39 | 84.13 | 87.50 | 90.45 | 94.16 | 82.05 | 81.62 | 79.60 |
>
> ---
>
> > **Q3.** **Alternative Sampling Metrics**: Comparing the effectiveness of evaluator uncertainty against alternative metrics would offer a more comprehensive justification.
>
> **A3.** Thanks for your constructive suggestions. In this work, our uncertainty-aware sampling uses **sampling-based entropy** to measure the uncertainty of the evaluator, which is inspired by self-consistency methods in LLM [1]. Besides the proposed metric, we also devise an alternative metric, **prediction-based entropy**, which uses greedy decoding to generate a single judgment of each training input, and computes the probability entropy of the judgment prediction. In the following table, we compare these two metrics with our URST. Sampling-based entropy consistently outperforms prediction-based entropy across all test sets.
>
> **Our sampling-based entropy uses temperature sampling and entropy of multiple outputs to measure the uncertainty of the evaluator, which could explore various reasoning paths for answer generation and provide a reliable assessment of the model’s confidence.** However, prediction-based entropy uses greedy decoding to generate a single output for entropy computation, and is prone to suffer from the over-confidence issue of LLM [2], thus making less reliable measurement.
>
> | URST | Total |  | AITW-ID-traj |  | AITW-OOD-traj |  | AW-OOD-traj |  |
> | --- | --- | --- | --- | --- | --- | --- | --- | --- |
> |  | Acc | F1 | Acc | F1 | Acc | F1 | Acc | F1 |
> | w/ prediction-based entropy | 84.67 | 81.07 | 86.67 | 89.47 | 91.67 | 73.68 | 79.82 | 75.68 |
> | w/ sampling-based entropy | 86.39 | 84.13 | 87.50 | 90.45 | 94.16 | 82.05 | 81.62 | 79.60 |
>
> [1] Self-Consistency Improves Chain of Thought Reasoning in Language Models
>
> [2] Can LLMs Express Their Uncertainty? An Empirical Evaluation of Confidence Elicitation in LLMs
>
> ---
>
> > **Q4.** **Baseline RL Methods**: There is no ablation or comparison with GRPO, PPO, or other standard reinforcement learning methods.
>
> **A4.** Thanks for your constructive suggestions. We would like to provide clarification regarding the comparison with GRPO. SGPO in URST is exactly GRPO without normalization in advantage computing. The experimental comparison of SGPO and GRPO is presented in row “URST” and “w/o normalization removing” of Table 2. **Compared to GRPO, SGPO can achieve an improvement of 1.44% in the overall F1 score.** We also show the detailed comparison in the following table.
>
> | URST | AITW-ID-traj | AITW-OOD-traj | AW-OOD-traj | Total |
> | --- | --- | --- | --- | --- |
> |  w/ GRPO | 89.61 | 78.95 | 77.95 | 82.69 |
> |  w/ SGPO | 90.45 | 82.05 | 79.60 | 84.13 |
>
> PPO needs an additional critic model for value computing, which requires a more complex training setting and a large dataset [1]. Recent studies [1,2,3] show that GRPO with verifiable rewards is computationally efficient and shows advantages in the CoT-based reasoning scenario, compared to PPO. Thus, we refer to GRPO as a strong RL baseline.
>
> [1] What’s Behind PPO’s Collapse in Long-CoT? Value Optimization Holds the Secret
>
> [2] VAPO: Efficient and Reliable Reinforcement Learning for Advanced Reasoning Tasks
>
> [3] DAPO: An Open-Source LLM Reinforcement Learning System at Scale
>
> ---
>
> > **Q5.** **Generalization Beyond Mobile GUI Tasks**: It would be helpful to test the pipeline on other GUI domains, such as desktop or web interfaces.
>
> **A5.** Thanks for your constructive suggestions. Recently proposed **AgentRewardBench** [1] is the first benchmark to **assess the effectiveness of MLLM-based evaluators for evaluating web agents.** It contains 1302 web trajectories across 5 benchmarks and 4 agents. Due to the lack of training data, we split AgentRewardBench into two sets, 1047 trajectories for the training set and 255 for the test set, and ensure a similar class distribution.
>
> |  | #Trajectory | #success | Success Rate |
> | --- | --- | --- | --- |
> | Training | 1047 | 293 | 27.99% |
> | Test | 255 | 63 | 24.71% |
>
> We evaluate the prompting-based (WebJudge, AutoEval*) and fine-tuning-based (SFT, URST) methods on this web benchmark, and report the results in the following Table. URST is trained under the setting of initial-300 samples, iter. 0-200 samples, iter. 1-200samples, and SFT is trained with the same total number of samples. Our URST achieves the highest performance on AgentRewardBench, reaching 87.06% accuracy and 77.85% F1 score, substantially outperforming the other baselines. **These comparisons underscore the effectiveness and generalization of URST.**
>
> | Methods | AgentRewardBench |  |
> | --- | --- | --- |
> |  | Acc. | F1 |
> | WebJudge | 83.07 | 68.15 |
> | AutoEval* | 85.10 | 74.67 |
> | SFT | 84.31 | 74.36 |
> | **URST** | **87.06** | **77.85** |
>
> [1] AGENTREWARDBENCH: Evaluating Automatic Evaluations of Web Agent Trajectories, arxiv 2025.

---

> ### Author Response · Authors · 2025-08-05
>
> Dear Reviewer ktZw,
>
> Thank you again for the valuable comments. Here is the brief summary of our rebuttal:
> - Concern about reliance on evaluator uncertainty: We provide clarification on this point, and conclude that the uncertainty serves as a useful signal for trajectory value assessment, and is unrelated to trajectory annotation quality.
> - Concern about evaluator evolution: We present detailed results across multiple iterations, and reveal a phenomenon of diminishing marginal returns.
> - Concern about alternative sampling metrics: we devise an alternative sampling metric, prediction-based entropy, and compare it with our originally proposed sampling-based entropy, providing a comprehensive justification of our original choice.
> - Concern about baseline RL methods: We conduct comparative experiments between GRPO and SGPO, demonstrating the superiority of our proposed SGPO.
>
> Other detailed responses can be found in the **Rebuttal**. As the interactive discussion window will close soon, we kindly invite you to review our responses and let us know if you have any further questions.
>
> Thank you!
>
> Best regards,
>
> Authors

---

> > ### Comment · Reviewer_ktZw · 2025-08-06
> >
> > Thank you for the detailed responses! I’m impressed by the consistent improvements across iterations. I’ll be raising my score accordingly.

---

> > > ### Author Response · Authors · 2025-08-06
> > >
> > > We sincerely thank Reviewer ktZw for the positive feedback and the decision to raise the score. We will include these additional experimental results and analyses in the revision. Thank you again.

---

### Official Review · Reviewer_3VNW · 2025-07-05

**Clarity:** 4
**Significance:** 3
**Originality:** 3
**Rating:** 5
**Confidence:** 4

**Summary:**

This paper proposes a novel framework called Uncertainty-aware Reinforced Self-Training (URST) for training lightweight multimodal large language models (MLLMs) to evaluate the quality of GUI navigation trajectories. Unlike existing methods that rely heavily on prompting proprietary models, URST trains evaluators using self-generated data, enhanced by uncertainty-aware sampling and simplified group policy optimization (SGPO). Extensive experiments demonstrate that URST outperforms state-of-the-art prompting-based and fine-tuning methods, particularly in generalization to out-of-domain data. The trained evaluator also enhances downstream GUI agents by filtering noisy training data.

**Questions:**

1. The paper reports performance metrics without error bars or statistical significance due to compute constraints. The authors could run multiple trials with different random seeds and report standard deviation or confidence intervals.

2. The reasoning performance is described qualitatively but not measured explicitly. It is suggested to quantify the quality of generated chain-of-thought (CoT) using human or model-based annotation. And add visualizations or examples of failure cases to show when/why URST succeeds or fails.

**Ethical Concerns:**

["NO or VERY MINOR ethics concerns only"]

**Final Justification:**

The author's respnse addressed the concerns/suggestions in my review. I have no other question.

**Limitations:**

yes

**Quality:**

3

**Strengths And Weaknesses:**

***Strengths***

First, the paper addresses a critical bottleneck in GUI agent training—noisy trajectory data—by proposing an efficient, cost-effective, and domain-adaptive evaluator. The use of lightweight open-source MLLMs and self-training avoids the high latency and cost of proprietary models. The URST framework incorporates two novel components: (1) uncertainty-aware sampling to prioritize informative, high-entropy samples and (2) SGPO to leverage both correct and incorrect samples for contrastive learning. These innovations are validated through rigorous ablation studies, showing measurable performance gains, especially in out-of-domain scenarios.

Second, the experiments are comprehensive and reproducible, with comparisons across multiple benchmarks, including in-domain (AITW-ID) and out-of-domain (AITW-OOD, AW-OOD) datasets. URST shows consistent improvements in F1 score and downstream agent performance. The authors clearly document their method, training configuration, datasets, and open-source code/data commitments, which greatly increases the impact and credibility of the work. Additionally, the qualitative examples illustrate robust reasoning capabilities and explain how URST corrects common failure modes in baseline methods.

***Weaknesses***

Despite its strengths, the paper relies on a proprietary MLLM to generate the initial training data, creating a partial dependency on the very systems it aims to replace. Although the authors argue this is limited to initialization, it could limit true independence in practice. Moreover, statistical significance and variability are not fully explored—the paper omits error bars due to compute limitations, which may raise questions about result stability.

Another minor weakness is that URST’s scalability to larger models or real-time deployment scenarios is not discussed in depth. While lightweight models are used for efficiency, more details on training cost, iteration limits, and potential bottlenecks (e.g., entropy computation) would strengthen the paper’s practical relevance.

Overall, this paper presents a timely and technically significant contribution to GUI agent training and trajectory evaluation by enabling efficient, domain-adaptive, and self-improving evaluators. The proposed URST method introduces meaningful algorithmic innovations and demonstrates solid empirical gains over existing approaches. Its potential for real-world impact is high, particularly for practitioners seeking to train robust GUI agents without relying heavily on commercial LLM APIs.

---

> ### Author Rebuttal · Authors · 2025-07-31
>
> We appreciate the reviewer’s insightful comments and helpful suggestions. Below are our responses and explanations:
>
> ---
>
> > **Q1.** The paper relies on a proprietary MLLM to generate the initial training data, which could limit true independence in practice.
>
> **A1.** We thank the reviewer for raising this important point about dependency on proprietary MLLMs. While our method does leverage a proprietary MLLM for both initial data generation *and* as a reward model during iterative self-training, this design reflects a pragmatic trade-off to address the absence of ground-truth trajectory annotations. Crucially, the role of the proprietary model diminishes over time: it is only used to score the judgment predictions, and the lightweight MLLM’s parameters are progressively refined using its own diverse CoTs. This mimics successful iterative approaches in other domains (e.g., robotics [1] or NLP [2,3]), where advanced models guide early-stage learning but are phased out as the student model converges.
>
> That said, we fully agree that reducing reliance on proprietary systems is desirable. However, our experiments suggest that **URST still advances independence compared to end-to-end reliance on proprietary MLLMs (like prompting-based methods, Webjudgment, AutoEval\*, and etc.)**, as the final evaluator operates without external API calls during deployment. We will add this limitation and direction to the revised version.
>
> [1] Self-Improving Robots: End-to-End Autonomous Visuomotor Reinforcement Learning, CoRL 2023.
>
> [2] Beyond Human Data: Scaling Self-Training for Problem-Solving with Language Models, Google DeepMind, TMLR 2024.
>
> [3] CREAM: Consistency Regularized Self-Rewarding Language Models, ICLR 2025
>
> ---
>
> > **Q2.** URST’s scalability or real-time deployment. More details on training cost, iteration limits.
>
> **A2.** Thanks for your constructive suggestions. In Table 1 of the main paper, we have presented URST with two different MLLMs, Qwen2VL-2B and Qwen2.5VL-3B. Compared to Qwen2Vl-2B, Qwen2.5VL-3B achieves a superior performance across all three test sets, indicating the scalability of URST.
>
> For real-time deployment, we analyze the inference time of prompting-based methods (WebJudge, AutoEval*) and fine-tuning-based methods (URST) in the following table. In terms of inference efficiency, URST demonstrates a significant advantage over both WebJudge and AutoEval. **URST achieves an average inference time of only 2.17 seconds per sample, which is approximately 17.6× faster than WebJudge (38.1 s) and 2.8× faster than AutoEval\* (5.99 s).** This substantial reduction in computational cost highlights the lightweight design of URST and its suitability for large-scale trajectory evaluation. We will include these results and analyses in the revised version.
>
> |  | WebJudge | AutoEval* | URST |
> | --- | --- | --- | --- |
> | Inference Time (s / per sample) | 38.1 | 5.99 | 2.17 |
>
> We also investigate the training cost of URST and provide detailed results on the training time for all iterations. All experiments were conducted on a server equipped with 4 A100 40GB GPUs. As shown in the table below, the initial phase with SFT requires only a small amount of time, whereas the subsequent iterative training with sampling and SGPO accounts for the majority of the time consumption. We will include these results and analyses in the revised version.
>
> |  | Initial | Iter. 0  | Iter. 1  | Iter. 2 |
> | --- | --- | --- | --- | --- |
> |  |  | Sampling / SGPO | Sampling / SGPO | Sampling / SGPO |
> | Training Time | 13min | 1h43min / 5h28min | 1h33min / 5h31min | 1h21min / 5h32min |
> | F1 | 77.88 | 79.43 | 82.85 | 84.13 |
>
> The table above also presents the performance across all iterations. It can be observed that each iteration consistently improves the evaluator’s performance. However, performing additional iterations (> Iter. 1) yields diminishing gains due to the relatively low quality of the newly sampled training data. Considering the computational cost, we limited our experiments to three iterations, which were sufficient to achieve the best performance observed in our study. Therefore, we did not further investigate the upper bound on the number of iterations. **We agree that the choice of iteration limits is an important factor, and we believe it can be determined by balancing training cost and performance requirements to reach an optimal trade-off under given computational constraints.** We will include these results and analyses in the revised version.
>
> ---
>
> > **Q3.** The authors could run multiple trials with different random seeds and report standard deviation or confidence intervals.
>
> **A3.** Thanks for your constructive suggestions. In the following Table, we present the statistical results of SFT and URST with three different random seeds. URST significantly outperforms SFT across all three test sets. These improvements are statistically meaningful given the non-overlapping confidence intervals, particularly for the F1 score, where URST shows a substantial gain (e.g., +4.82 on AITW-OOD-traj). Moreover, the smaller standard deviations in most metrics indicate that URST produces more stable results across different runs. **Overall, these findings demonstrate that URST not only improves performance but also enhances result consistency.**
>
> | Methods | AITW-ID-traj |  | AITW-OOD-traj |  | AW-OOD-traj |  | Overall |  |
> | --- | --- | --- | --- | --- | --- | --- | --- | --- |
> |  | acc | f1 | acc | f1 | acc | f1 | acc | f1 |
> | SFT | 83.61±0.79 | 87.15±0.50 | 93.61±1.42 | 78.71±3.00 | 77.13±0.97 | 71.30±1.17 | 83.08±0.71 | 78.62±0.71 |
> | URST | 87.78±0.39 | 90.64±0.27 | 94.72±0.78 | 83.53±2.09 | 83.40±1.27 | 82.14±1.80 | 87.47±0.81 | 85.59±1.05 |
>
> ---
>
> > **Q4.** It is suggested to quantify the quality of generated chain-of-thought (CoT). And add visualizations or examples of failure cases.
>
> **A4.** Thanks for your constructive suggestions. To evaluate the reasoning quality of different models, we conducted pairwise comparisons of their Chain-of-Thought (CoT) outputs by using GPT-4.1. Specifically, URST was compared separately with SFT and AutoEval*, and the winning rates of the latter models were recorded. As shown in the table, SFT outperformed URST in only **32%** of the paired cases, while AutoEval* surpassed URST in only **36%** of the comparisons. These results indicate that **URST consistently maintains a clear advantage over both baselines**, as it was judged superior in the majority of cases across both pairwise evaluations. This suggests that **URST’s reasoning process is more reliable and better aligned with human preferences than that of the other models.**
>
> |  | SFT is winner | AutoEval* is winner |
> | --- | --- | --- |
> | URST | 32% | 36% |
>
> In section 4.5 and C.2, we have presented the case study of examples where URST succeeds or fails. Here, we would like to restate interesting and insightful findings:
>
> 1. When inspecting the success cases of URST, we found that URST has advantages of discerning subtle differences, mitigating hallucinations, and tracking fine-grained transitions (as shown in Figure 5 of the main paper and Figure 2 of the Appendix).
> 2. When inspecting the failure cases of URST, we found that URST has limitations of insufficient observation, GUI elements misinterpretation, and instruction misunderstanding (as shown in Figure 3 of the Appendix).

---

> ### Author Response · Authors · 2025-08-05
>
> Dear Reviewer 3VNW,
>
> Thank you again for the valuable comments. As the interactive discussion window will close soon, we kindly invite you to read our responses and let us know if you have any further questions.
>
> Thank you!
>
> Best regards,
>
> Authors

---

### Note · Authors · 2025-08-12

We sincerely thank the reviewers for their time, thoughtful feedback, and constructive discussion. During the discussion phase, we were pleased to have largely addressed their concerns.

For reviewer **3VNW**, we highlighted the independence of URST in practice, provided a detailed analysis of its training and inference costs, and examined the quality of the generated CoTs. We sincerely thank the reviewer for the consistent acceptance decision.

For reviewer **ktZw**, we clarified the value of uncertainty measurement, conducted a detailed analysis of evaluator evolution, compared the proposed uncertainty-aware sampling and SGPO with diverse alternatives, and extended URST to a Web GUI benchmark. We appreciate the reviewer’s positive feedback and the decision to raise the score.

For reviewer **oZh1**, we clarified the change in entropy, emphasized the significant improvements achieved by URST, and evaluated it on an LLM agent benchmark (*JudgeBench*), demonstrating its remarkable generalization. We sincerely thank the reviewer for the positive feedback and score.

For reviewer **nyzi**, we clarified the soundness and comprehensiveness of the baseline settings, and highlighted the three key roles of the trajectory evaluator with convincing experiments. We appreciate the reviewer’s positive feedback and the decision to raise the score.

Overall, we present an uncertainty-aware reinforced self-training method to train an evaluator for autonomously assessing trajectories in GUI environments. This evaluator also functions as a powerful filter for cleaning noisy GUI navigation datasets collected under diverse conditions and provides a valuable reward signal that enhances GUI agent inference. We establish comprehensive baselines and demonstrate the superiority of the proposed uncertainty-aware sampling and SGPO techniques. Our URST method exhibits remarkable generalization, validated not only on multiple Mobile GUI benchmarks under both in-domain and out-of-domain settings, but also extended to a Web GUI benchmark (*AgentRewardBench*) and an LLM agent benchmark (*JudgeBench*).

---

### Decision · Program_Chairs · 2025-09-17

**Decision:**

Accept (poster)

**Comment:**

The paper received all positive reviews, leading to a final acceptance recommendation.